# Spatially resolved gene expression profiling of tumor microenvironment reveals key steps of lung adenocarcinoma development

Yuma Takano[1,2], Jun Suzuki[3,4], Kotaro Nomura[3], Gento Fujii[1], Junko Zenkoh[1], Hitomi Kawai [5], Yuta Kuze[1], Yukie Kashima[1], Satoi Nagasawa[1], Yuka Nakamura [6], Motohiro Kojima[6], Katsuya Tsuchihara [7], Masahide Seki [1], Akinori Kanai [1], Daisuke Matsubara[5], Takashi Kohno [8], Masayuki Noguchi[5,9], Akihiro Nakaya [1], Masahiro Tsuboi[3], Genichiro Ishii[6], Yutaka Suzuki [1] ✉ & Ayako Suzuki [1] ✉

The interaction of tumor cells and their microenvironment is thought to be a key factor in tumor development. We present spatial RNA profiles obtained from 30 lung adenocarcinoma patients at the non-invasive and later invasive stages. We use spatial transcriptome sequencing data in conjunction with in situ RNA profiling to conduct higher resolution analyses. The detailed examination of each case, as well as the subsequent computational analyses based on the observed diverse profiles, reveals that significant changes in the phenotypic appearances of tumor cells are frequently associated with changes in immune cell features. The phenomenon coincides with the induction of a series of cellular expression programs that enable tumor cells to transform and break through the immune cell barrier, allowing them to progress further. The study shows how lung tumors develop through interaction in their microenvironments.

Tumor cells can exhibit significant transcriptomic diversity depending on their location and environment. Tumor cells are constantly exposed to various stresses due to their environment, which is made up of various types of cells, such as tumor-infiltrating leukocytes and cancer-associated fibroblasts (CAFs)[1–3]. The human body eliminates tumor cells through immune cell attack[4], and in fact, a local immune cell structure known as tertiary lymphoid structures (TLSs)[5,6], consisting of cytotoxic T cells (CTLs), activated B cells, and other immune cells, is observed in tumor tissues. These TLSs are thought to act as a local supply base for immune cells targeting tumor cells. However, tumor cells survive these adverse events by adapting to a niche in the local tumor microenvironment (TME) through changes in their transcriptional programs. For example, transcriptional changes in a group of genes that induce tolerance to immune cells have been reported[7]. The increased physical plasticity of tumor cells allows them to pass through the physically paved barriers of normal cells and escape into larger spaces. Tumor cells also develop physiological plasticity as they progress, which means they rely less on oxygen or nutrition supplies,

[1]Department of Computational Biology and Medical Sciences, Graduate School of Frontier Sciences, The University of Tokyo, Kashiwa, Chiba, Japan. [2]Pharmaceutical Science Department, Chugai Pharmaceutical Co., Ltd., Chuo-ku, Tokyo, Japan. [3]Department of Thoracic Surgery, National Cancer Center Hospital East, Kashiwa, Chiba, Japan. [4]Department of General Thoracic Surgery, Juntendo University Graduate School of Medicine, Bunkyo-ku, Tokyo, Japan. [5]Department of Diagnostic Pathology, Faculty of Medicine, University of Tsukuba, Tsukuba, Ibaraki, Japan. [6]Division of Pathology, Exploratory Oncology Research and Clinical Trial Center, National Cancer Center, Kashiwa, Chiba, Japan. [7]Division of Translational Informatics, Exploratory Oncology Research and Clinical Trial Center, National Cancer Center, Kashiwa, Chiba, Japan. [8]Division of Genome Biology, National Cancer Center Research Institute, Chuo-ku, Tokyo, Japan. [9]Center for Clinical and Translational Science, Shonan Kamakura General Hospital, Kamakura, Kanagawa, Japan. ✉e-mail: ysuzuki@edu.k.u-tokyo.ac.jp; asuzuki@edu.k.u-tokyo.ac.jp

**Table 1 | General information of the eight IA cases**

| Case | Smoking history | Pathological stage | Histology[a] | Histological predominant[b] | Driver mutation | Spatial omics analysis | | |
|---|---|---|---|---|---|---|---|---|
| | | | | | | Visium | PhenoCycler | Xenium |
| LUAD No. 1 | Never | IA3 | M/D | Papillary | *EGFR* ex19del | FF | *** | *** |
| LUAD No. 2 | Former | IIB | M/D[c] | Lepidic | *KRAS* G12A | FF/FFPE | FFPE | FFPE |
| LUAD No. 3 | Never | IA3 | M/D | Papillary | *EGFR* ex19del | FF/FFPE | FFPE | FFPE |
| LUAD No. 4 | Never | IB | M/D | Papillary | *EGFR* L858R | FF/FFPE | FFPE | *** |
| LUAD No. 5 | Never | IIB | M/D | Papillary | *EGFR* L858R | FF | *** | *** |
| LUAD No. 14 | Current | IVA | M/D | Acinar | *KRAS* G12A | FF | *** | FF |
| LUAD No. 16 | Former | IA3 | M/D | Papillary | *EGFR* ex19del | FF | *** | FF |
| LUAD No. 17 | Never | IIIA | P/D | Solid | *EGFR* L858R | FF | *** | FF |

[a]M/D: moderately differentiated. P/D: poorly differentiated.
[b]Histological predominance was evaluated in the initial pathological diagnosis for each case.
[c]LUAD No. 2: mucinous > 10%.
***: Not performed.

making them more suitable for further expansion and metastasis[8]. Such events are frequently discussed in conjunction with a feature of epithelial–mesenchymal transition (EMT)[9]. In fact, tumor cells that underwent EMT exhibit therapeutic resistance, metastasis, and a poor prognosis. However, our understanding of the actual molecular and cellular events that drive these changes during human tumor progression, such as interactions between tumor and microenvironmental cells, is limited. This stands in stark contrast to the fact that extensive genomic analyses have revealed key mutational events in a variety of malignancies. Such knowledge would allow us to eventually eradicate cancer by inducing beneficial molecular and cellular changes in individual tumor tissues.

Here, to investigate molecular and cellular events during tumor progression, we performed a large-scale spatial transcriptome analysis[10–12] of 30 lung tumors, including 22 early non-invasive tumors including adenocarcinoma in situ (AIS) and minimally invasive adenocarcinoma (MIA) and eight late invasive tumors. First, the local transcriptome features of tumor cells, environmental stromal cells, and infiltrating immune cells were examined for invasive adenocarcinomas (IA) by the Visium (10x Genomics)[10] examination with a 55 μm resolution. Interactive events for some representative samples observed at the boundaries between malignant (i.e., invasive) frontier and less malignant regions were further examined at single-cell level resolution using a PhenoCycler multiplexed immunostaining[13] and Xenium in situ gene expression profiling[14,15] systems. The interactive events between tumor and immune cells, as well as phenotypic factors thought to be important for disease progression, were then investigated in lung tumors at earlier stages, such as AIS and MIA. These studies identified TME surrounding tumor cells as an important event associated with disease progression in lung adeno-carcinogenesis.

## Results
### Overview of study design
In this study, spatial omics analyses were performed on 30 lung adenocarcinoma cases (eight IA and 22 AIS/MIA cases). The datasets used in this study are listed in Supplementary Table 1. Genetic and clinicopathological information for IA cases is summarized in Table 1. For the AIS/MIA cases, we published genomic statuses and clinicopathological information recently[16] (also see Supplementary Table S1).

The overall workflow for this study is depicted in Fig. 1. Spatial transcriptome sequencing[10] Visium was performed on 16 tissue sections from eight IA cases with *EGFR* or *KRAS* driver mutations (Supplementary Table 2). We then analyzed Visium datasets from 28 sections of 22 AIS/MIA cases, 25 of which were newly obtained (Supplementary Table S3). Both fresh frozen (FF) and formalin-fixed and paraffin-embedded (FFPE) specimens were utilized depending on the samples. For the higher resolution analysis, PhenoCycler multiplexed

immunostaining was performed on representative cases (Supplementary Tables S4 and S5). Finally, spatial expression profiling Xenium was conducted, and 17 sections from five IA and 12 AIS/MIA cases (two from our previous study[16]) were analyzed (Supplementary Tables S6 and S7). Using Visium data from IA cases (Supplementary Fig. S1; Supplementary Fig. S2 for PhenoCycler data), we first examined the observed diverse expression profiles for several individual cases. More specifically, we focused on LUAD No. 2 and 3, which harbored *KRAS* or *EGFR* driver mutations, respectively. *EGFR* and *KRAS* are two of the most important driver genes in lung adenocarcinoma, accounting for a large proportion of cases. We used data from FFPE sections that included regions with transcriptomic/histological features of both well-differentiated and malignant regions, which is critical for understanding tumor cell and microenvironmental characteristics in boundary regions. Then, we attempted to generalize the factors that would determine the fate of the cancers.

### Spatial expansion patterns of LUAD No. 2 (KRAS, mucinous cancer)
First, we examined one case, LUAD No. 2, as an example, before proceeding more general analysis. This case was selected as a representative IA case because it harbors a common driver *KRAS* mutation (Table 1). In this case, Visium spots were divided into 11 clusters (FFPE section C, Fig. 2a; information on all sections of LUAD No. 2 is depicted in Supplementary Fig. S3). Normal or well-differentiated regions (clusters 1, 3, 4, and 5) were identified by generally high expression of *NAPSA* and surfactant genes[17,18], indicating that these regions possessed normal alveolar epithelial cell characteristics (Fig. 2b and Supplementary Fig. S4b). Notably, cluster 3 showed high *DUOX1* expression (Supplementary Fig. S4c), indicating that tumor cells in these regions are subjected to oxidative stress[19]. Cluster 6 contained an inflammatory region with high levels of *IFITM1* expression. Mucinous regions exhibited high *MUC5AC* expression[20] in clusters 0 and 8. Clusters 4 and 9 were immune cell-rich regions, whereas clusters 2, 9, and 10 had *COL1A1*, *ACTA2*, and *SPARC* expression, which are fibroblast and CAF markers (Supplementary Fig. S4c and d).

We attempted to identify the key factors that influence these patterns. We examined changes in gene expression at the cluster boundaries. In this case, as well as other study cases described below, tumor cells showed drastic changes in gene expression patterns as they crossed the borders defined by regions of immune cells, which appeared as a "stream."

In LUAD No. 2, pathological examination roughly divided the cancer into two parts: 1) the left side, which represented normal or less malignant well-differentiated tumor cells, and 2) the right side, which represented more malignant or mucinous regions (Supplementary Fig. S4a). Accordingly, the Visium data revealed high expression of

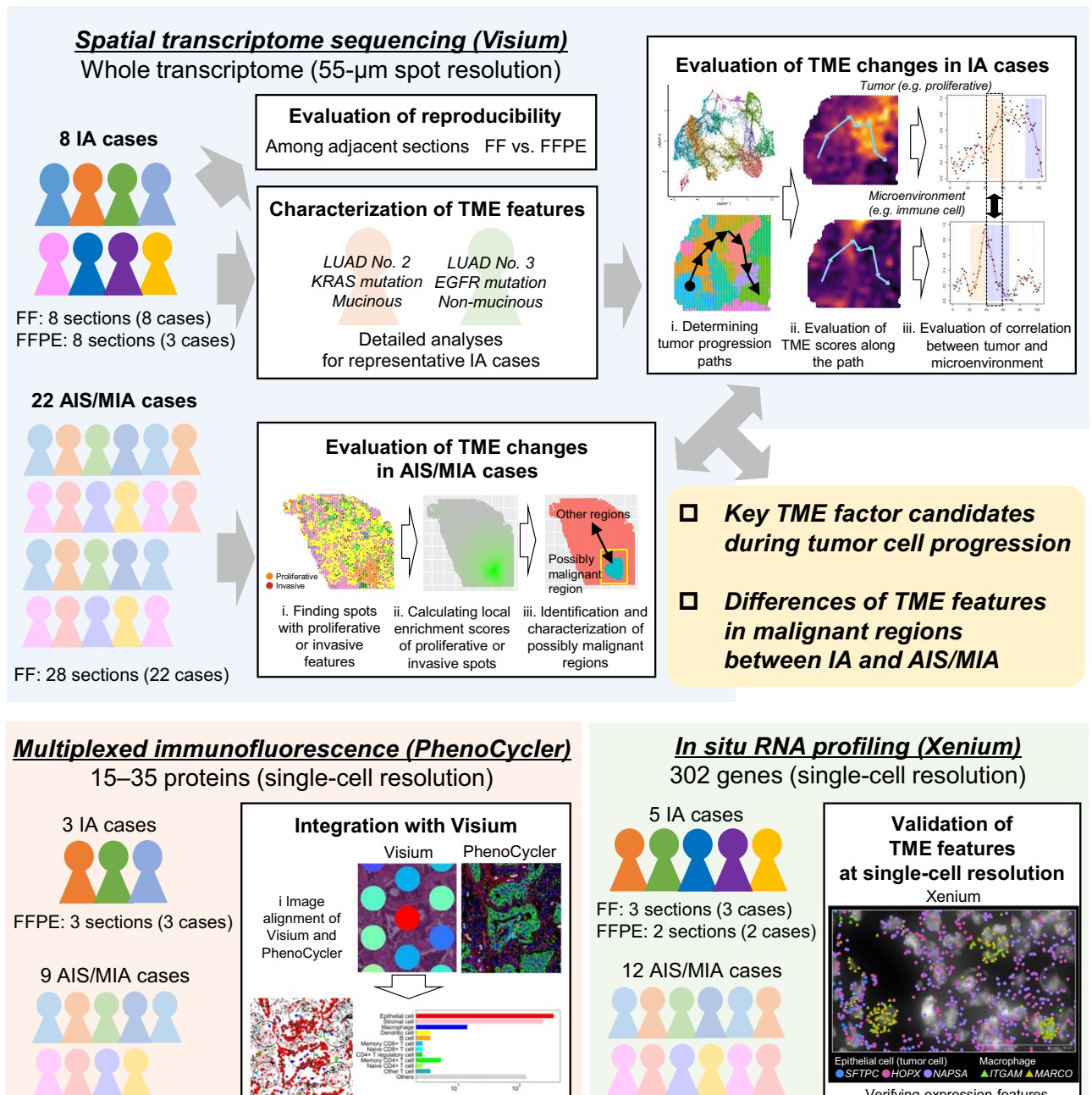

**Fig. 1 | Overview of the study.** The overall analytical workflow including sample information and the spatial omics platforms used in this study.

typical differentiated markers, such as *NAPSA*, on the left side of the clusters (Fig. 2b). In this region, a vital transcription factor of alveolar epithelium lineages, *NKX2-1*, was active (left, Fig. 2c). In contrast, in the right region, *NKX2-1* expression was lost, while another transcription factor, *HNF4A*, was active (right, Fig. 2c). HNF4A is a key transcription factor in gastrointestinal epithelium, which suggests that tumor cells may transdifferentiate in this region[21,22]. Consistently, mucin production began, as evidenced by the expressions *CDX2* and *MUC2* (region 3, Supplementary Fig. S4a).

A closer look at this boundary region revealed the presence of a cluster labeled "active inflammatory reactions" (cluster 6). Here, typical inflammatory response genes, such as *IFITM1* and *IFI6*, were found to be highly activated (Fig. 2d) and associated with interferon signaling (Fig. 2e). Adjacent to this region, we discovered an immune cell-rich cluster (cluster 4) near a lymphoid follicle structure (region 5,

Supplementary Fig. S4a), implying extensive immune cell infiltration and attack in this region as a result or cause of the tumor cell's drastic transition. In this boundary region, tumor cells activated *IDO1* (cluster 6; log2 fold change = 0.86, adjusted *p*-value = 2.0e-65). *IDO1* activation was demonstrated at the protein level (Fig. 2f). IDO1 is induced by IFN-γ (type II interferon) and suppresses effector T and NK cells while activating regulatory T cells (Tregs) and myeloid-derived suppressor cells[23].

As they progressed to the right end region, tumor cells developed new transcriptional and phenotypic characteristics. In the end region, protective mucin expression was no longer detected (Fig. 2g), and tumor cells exhibited high proliferative markers. In their surrounding stroma, CAF markers *ACTA2* and *SPARC* were found to be highly expressed (Fig. 2h and i). When we looked at the boundary between the mucinous region and the CAF-rich invasive region (cluster 2), we discovered high expression of genes associated with extracellular matrix

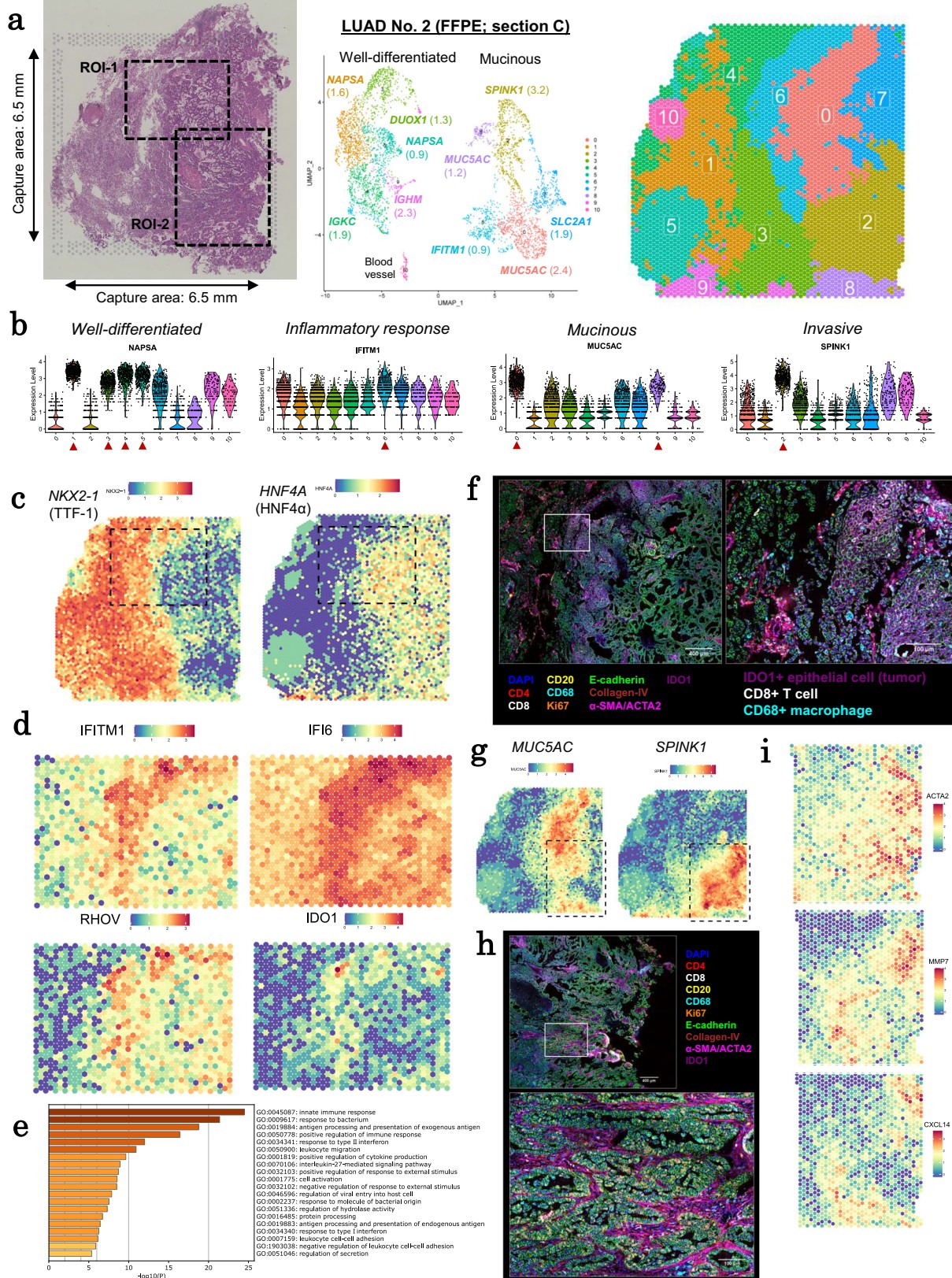

remodeling and anoikis resistance, such as *SPINK1*[24,25] and *MMP7*[26–28] (Fig. 2g and i).

Collectively, it was proposed that tumor cells in the initial boundary region (cluster 6 between clusters 4 and 0) had begun to develop inhibitory responses to immune cells. It is possible that the mucin expression was induced to protect tumor cells from immune cell attacks. Furthermore, it was discovered that tumor cells in this region began to evolve into a more malignant state. In the second boundary (clusters 0 and 2), tumor cells were surrounded by a fibroblast blanket, and immune cell attacks were significantly reduced. Furthermore, *CXCL14*, an invasion-associated factor[29,30], with an unknown role in the TME[31], was overexpressed in this region. At this

**Fig. 2 | Spatial transcriptome analysis of LUAD No. 2. a** H&E (left) and results of clustering analysis (middle and right) of LUAD No. 2 FFPE section C. The H&E image shows two regions of interest (ROIs). The capture area that is surrounded by the fiducial frame in the H&E image is 6.5 mm × 6.5 mm. **b** Violin plots of marker genes in each cluster. The plots for some other markers are also depicted in Supplementary Fig. S4c. **c** Spatial distribution of the expression levels of *NKX2-1* (left) and *HNF4A* (right). The ROI-1 is represented as a dashed square. **d** Gene expression levels in the boundary region (ROI-1). **e** Gene set enrichment analysis on genes that were highly expressed in cluster 6 using Metascape (version 3.5)[58].

The *p*-values were calculated based on the hypergeometric test by Metascape. **f** Expression of IDO1 and representative cell markers in PhenoCycler immunostaining. Left: ROI-1. Right: an enlarged view of the white square in the left panel. **g** The spatial distribution of the expression levels of *MUC5AC* and *SPINK1*. The ROI-2 is represented as a dashed square. **h** PhenoCycler immunostaining in the mucin-negative, invasive area. Top: ROI-2. Bottom: an enlarged view of the white square in the top panel. **i** Expression of CAF markers and representative DEGs in the mucin-negative area (ROI-2). Source data are provided as a Source Data file for (**e**).

stage, protection from mucin may not be as beneficial to tumor cells as it was in the less-invasive stages, particularly for expansion.

### Spatial expression of LUAD No. 3 (EGFR, non-mucinous cancer)

We looked LUAD No. 3 (FFPE section B) for another example (Fig. 3a and b). This case was chosen as an example of cases having another common *EGFR* mutation (Table 1). The pathological diagnosis revealed a more complicated regional structure, so this was chosen as the case. Pathologically, this tissue predominantly showed a papillary pattern. The tumor cells are thought to have originated in clusters 1 or 9. During their expansion from this region to other regions, such as clusters 6 and 3 regions, the tumor cells exhibited a more malignant expression pattern, such as high expression levels of malignant markers, such as *TNC* and *TGFBI* (Supplementary Fig. S5a and b), as well as active proliferation markers, such as *FOS* and *WEE1* (Supplementary Fig. S5c). EMT markers like *RHOB* and *VIM*, were also upregulated in cluster 3 (Fig. 3c). Between clusters 6 and 3, a large immune cell-rich region (cluster 5) was discovered (Fig. 3d). The region contains various immune cell types, such as B cells, T cells, and macrophages, in this region. However, we discovered that resolution in Visium analysis was insufficient to analyze the detailed distribution patterns of individual immune cells and their interactions with tumor cells.

To obtain finer-resolution images, we used multiplexed immunostaining analysis with PhenoCycler at least for genes of interest. Because the vertically consecutive sections from Visium analysis needed to be used for PhenoCycler multiplexed immunostaining analysis, we created a bioinformatics pipeline to superimpose images from Visium and PhenoCycler (Supplementary Fig. S6; detailed description is provided in the "Methods" section). The PhenoCycler analysis enabled highly resolution spatial analysis (Fig. 3e and Supplementary Fig. S6).

By combining Visium and PhenoCycler data, we were able to investigate the immune cell distribution pattern in cluster 5. We discovered that the characteristic immune cells did not always to have a uniform distribution even in this region. The expression of exhausted T cells increased in the left peripheral region of cluster 5 ("Downstream" in Fig. 3e), which is consistent with the findings of the PhenoCycler analysis of FOXP3 + CD4 T cells (Supplementary Fig. S5e). This lends support to the notion that the TME of the corresponding part of cluster 5 should reflect more exhausted features of immune statuses. In contrast, in central ("upstream") regions of the left side, the distribution of active cytotoxic immune cell populations (CTLs, plasma cells, etc.) remained relevant. Consistently, tumor cell presence was more relevant in downstream regions than in upstream areas.

CXCL13+ immune cells co-localized with CXCR5+ cells in cluster 6, which spread from cluster 5's immune cell region (Fig. 3f). Tumor-associated macrophages (TAMs) that express *MMP9* and *APOE* were also present in this cluster. Toward the nearest region to cluster 3, CTL markers, such as *PRF1*, *GZMA*, and *GZMK*, were highly expressed. However, they did not invade cluster 3's internal region (Supplementary Fig. S5d). Macrophages with high *SPP1* expression, which reportedly play an important role as anti-inflammatory TAMs involved in angiogenesis[32], were found in the region of cluster 3 (Fig. 3g and

Supplementary Fig. S5f and h). These macrophages expressed both the M2 macrophage marker *CD163* and the alveolar macrophage marker *MARCO*. SPP1+ macrophages may interact with CAFs[33] and induce EMT[34]. Consistently, fibroblast markers, such as *ACTA2*, were elevated in cluster 3 (Fig. 3g).

Taken together, we believe the following scenarios could shape the cancer landscape. First, the tumor cells crossed the barrier at the downstream part of cluster 5 and spread into the region of cluster 3. Prior to this potential expansion, in the region of cluster 6, immune cell subsets, such as CXCL13+ lymphocytes and TAMs, were important. These cells are thought to be associated with the pro-inflammatory microenvironment, which may trigger an effective response to PD-L1 blockade[35]. Therefore, tumor cells and TME with immune cell infiltration in cluster 6 should continue to respond to immune checkpoint blockade (ICB). Once cancer cells had spread to the regions of cluster 3, interactions between SPP1+ macrophages and fibroblasts were observed, which may aid in avoiding cytotoxicity by immune cells and, as a result, EMT in tumor cells themselves. When cancer tissue reaches this stage, it is more likely that ICB efficacy will be limited. Such heterogeneity in tumor cell status and TME may contribute to patients' overall ineffective response and resistance to immunotherapies. The inferred molecular features of TMEs in clusters 3, 5, and 6, are summarized in Fig. 3h.

### Diverse TMEs and tumor cells in the same block of LUAD No. 3

On the other side of the same section in LUAD No. 3, we discovered even more diverse microenvironment statuses developed. The region of cluster 11 did not contain any tumor cells. Predominant naïve immune cell markers, such as *CCR7*, were found instead (Supplementary Fig. S5g). Gaps were discovered in several locations where tumor cells were likely to have originated, implying that cancer cells that migrated in this direction were killed by immune cells.

Cluster 10 contained the cancer's most invasive and potentially fatal component. Similar to clusters 3 and 6, a stream of immune cells (cluster 5) was found in the peripheral region of cluster 10. In this case, there was no clear separation between exhausted immune cells (Supplementary Fig. S5g). *ACTA2* was highly expressed in this region, indicating that the tumor cells were closely linked to myofibroblasts and CAFs (Fig. 3i). Furthermore, we found high levels of a number of relevant matrix metalloproteinases, including *MMP7* and *MMP11*, in the cluster 10 region (Fig. 3i and j). Immune cells were almost completely excluded from this area. For this region, we believe that when tumor cells spread across the fibroblast zone, they may have developed the ability to control interactions with fibroblasts. This ability would be further used to exclude immune cells from tumor cells, potentially allowing the cancer to spread further.

Among all regions, the most notable discussion should be as follows. In the current scenario, the chemical therapeutic treatment for this patient is either a tyrosine kinase inhibitor or an ICB. However, this decision is being made, primarily based on the genomic mutation information, despite the fact that their expected effects vary even within this small section (6.5 mm square in width). To develop therapeutic strategies for such cases, it may be better to consider the transcriptomic features and their heterogeneity.

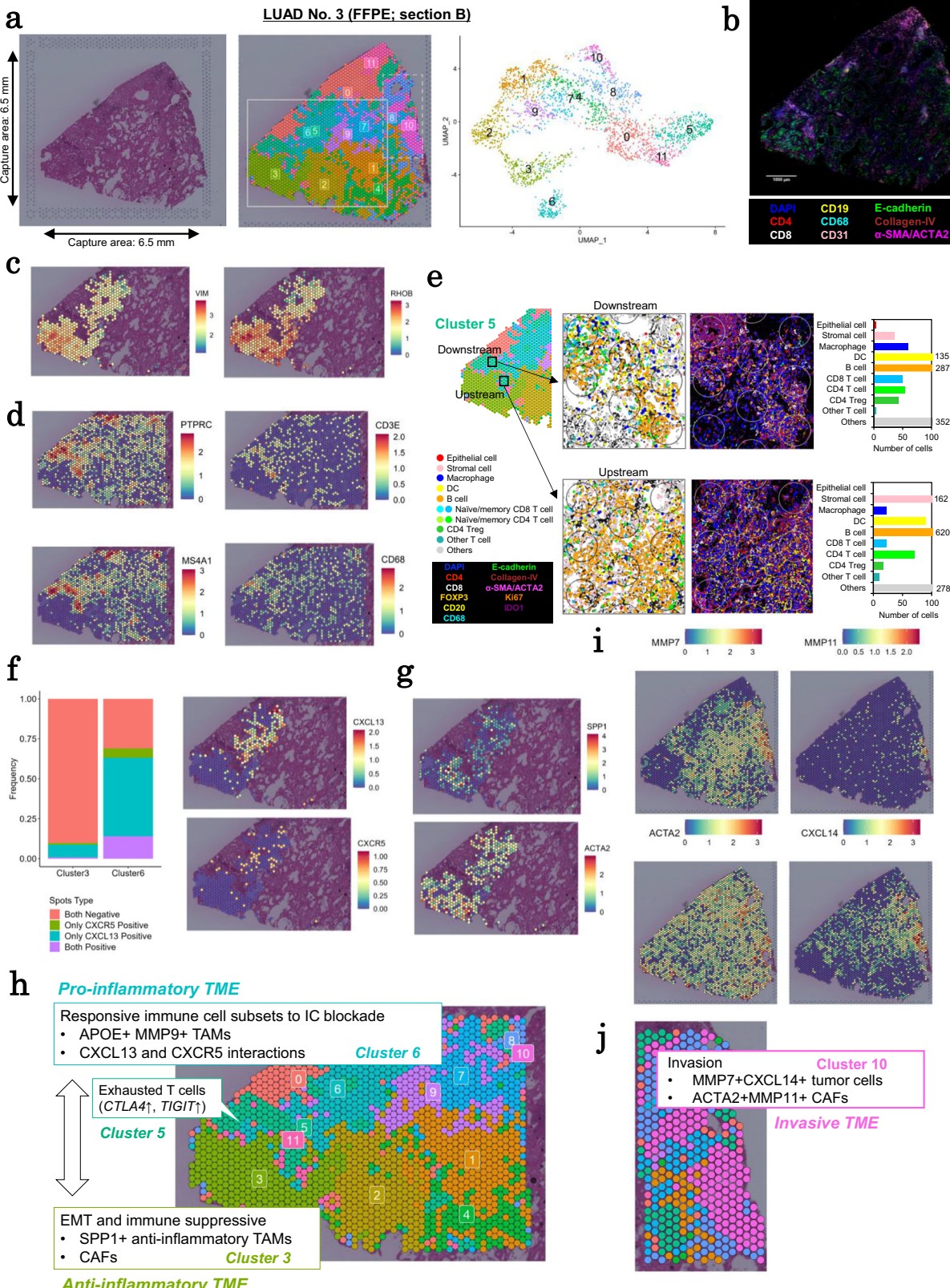

## Cross-case or section scoring of tumor cells and surrounding cells in IA cases

After examining several individual cases, we sought to generalize spatially distinctive gene expression profiles and their mutual association from a broader perspective. Because upregulated individual genes can vary by patient, we focused on gene groups commonly used to represent transcriptomic statuses, such as differentiation, proliferation, invasion, and immune cell activation or repression. Genes were chosen based on differences in expression patterns between the clusters and other studies[17,18]. The spatial expression patterns of the selected genes were converted to activity scores using PAGE analysis of Giotto[36] (Supplementary Table S8). The results were manually

**Fig. 3 | Local transitions to invasive phenotypes in LUAD No. 3. a** H&E (left), Visium clustering (middle), and UMAP visualization of the clustering results (right) of LUAD No. 3 FFPE section B. The capture area that is surrounded by the fiducial frame in the H&E is 6.5 mm × 6.5 mm. The spatial plot of the clustering result shows two ROIs. **b** A representative image of PhenoCycler immunostaining. **c** Differential expression of EMT-related genes in clusters 3 and 6. **d** Expression patterns of representative immune cell markers in the immune cell cluster 5. **e** Integrating Visium and PhenoCycler. Two ROIs in cluster 5 and color legends are shown (left). Cell types determined using PhenoCycler data and signals are shown for ROIs (middle). The number of cells detected in PhenoCycler data is shown for each ROI (right). **f** Expression distribution of CXCL13–CXCR5 pairs in clusters 3 and 6. The bar plot shows the proportion of spots classified based on CXCL13 and CXCR5 expression patterns. **g** Expression of *SPP1* (macrophages) and *ACTA2* (myofibroblast/CAFs) in clusters 3 and 6. **h** A schematic representation of the molecular characteristics of tumor clusters 3 and 6, as well as immune cell cluster 5, in LUAD No. 3. **i** Patterns of highly expressed genes in cluster 10. **j** A schematic representation of the molecular characteristics of cluster 10 in LUAD No. 3. Source data are provided as a Source Data file for (**e** and **f**).

checked, and the overall consistency of the annotations obtained from Visium clustering analysis was confirmed. Figure 4a depicts some example statuses in FFPE section C of LUAD No. 2 (refer to Supplementary Fig. S7 for full images). Using this scoring method, mutual comparisons were carried out between different specimens.

Based on the calculated scores, each spot was classified for mutual comparison. For IA cases, the areas of a given status could be depicted more simply than the expression profiles themselves (Fig. 4b). By comparing each score, status degrees between specimens can be compared. Based on the obtained data, we could consistently compare the width of the area with a given status between specimens. For example, the FF section of LUAD No. 2 had the largest area of well-differentiated cells (an adjacent region located on the left side of FFPE section C of LUAD No. 2, which was evaluated above). For immune cells (including both B and T cells), the most prominent area was in FFPE section A of LUAD No. 3. The score distribution for each status for each section is presented in Fig. 4c. All the results were consistent with results of the manual inspection by the pathologists, including the inter-case comparisons.

We concentrated on invasive areas (red color) as the most malignant feature in each section. In LUAD No. 14, the largest invasive area had the most poorly differentiated phenotype (Fig. 4d). FFPE section B of LUAD No. 3 had an invasive area on the right side of the section (Fig. 4e). In this region, cells other than CAFs expressed invasion-related genes. Notably, no clear histological changes were observed in this region, implying that transcriptomic profile-based scoring can be more useful for dissecting features that are obscured by limited histological or morphological information used for pathological examination.

Using the generated scoring scheme for local profiling, we attempted to determine how each profile might be related to the other. Before systematically comparing the profiles in the following section, we manually examined the possible association between the features in various regions of several specimens (Fig. 4b). We discovered several possible associations. For example, we found that the M2 macrophage profile was associated with the regions of tumor cells, particularly those with an aggressive/proliferative profile. Certain types of macrophages, such as anti-inflammatory macrophages, may have helped to create a favorable microenvironment for tumor cell proliferation and expansion while also excluding other types of immune cells, such as B cells and CTLs.

## Mutual associations of the local profiles

To better understand the molecular signatures of tumor cells adapting to their respective microenvironments, we incorporated the inferred route of cancer progression, identified through Visium trajectory analysis, into the above TME scoring (Fig. 5a and b; also see Supplementary Fig. S8a). We determined how the cancer should have moved across the TME landscape (Fig. 5c). Furthermore, we identified profiles that showed mutually positive or negative correlations with changes in tumor cell profiles. For the representative cases with heterogeneous of tumor cell clusters, a positive correlation was found between malignant and CAF profiles (Fig. 5d). The negative association between invasion and immune cells was the most significant (Fig. 5d). Relevant changes in tumor cells affecting

profile landscapes, such as drastic transformations of tumor cells or changes in pathological phenotypes, occurred in the overlapping area of the peak region of immune cell activity. The interaction was clear in at least four cases (eight transcriptome trajectories), depending on the threshold (Fig. 5e).

Given all of the data generated and analyzed for various IA specimens, we highly believe that immune cells may act as one of the most significant barriers to tumor cell expansion. These tumor cells can only grow if they undergo phenotypic changes in response to an immune cell attack.

## Validation analyses of TME at the single-cell level in IA cases

We aimed to validate expression profiles, also known as profile landscapes, and their mutual interactions at the single-cell level. For this purpose, we performed in situ gene expression analysis with Xenium (Fig. 6) on five specimens dissected from the same tissue blocks used in Visium analysis. Although data were only obtained from 302 designated genes, single-cell resolution data were obtained for all individual cells in the section, with an average of 264,710 cells per section. Using clustering analysis of the obtained Xenium data, we could classify each cell distinguishing between stromal and immune cells and tumor cells (Fig. 6a and Supplementary Fig. S9b). Furthermore, Xenium single-cell expression patterns allowed us to decompose Visium data at 55 μm resolution (Supplementary Fig. S9c).

We used the obtained data to validate the transcriptomic characteristics of tumor cells, with a focus on cell lineage and differentiation markers. The Xenium data clearly showed a representative profile change--from *NKX2-1*- to *HNF4A*-positive tumor cells in LUAD No. 2 (Fig. 6b). Xenium's finer resolution analysis revealed that cells expressing *NKX2-1* or *HNF4A* were adjacent to each other in a mutually exclusive way (Fig. 6c). Furthermore, the immune response and high *IDO1* expression were found to be relevant. We also found CCL22-positive cells, which could be DC or T cells, in this region. This feature was first discovered by deconvolution analysis of Visium data using Xenium data (Fig. 6d). CCL22 is a chemokine that attracts Tregs and promotes immune suppression[37]. In conjunction with IDO1 expression on the tumor cell side, CCL22-positive cells create local environments that protect tumor cells from immune cell attacks. Consistently, single-cell resolution Xenium data revealed that at least several CD8 + T cells were infiltrating and interacting, potentially attacking tumor cells in this region. Tumor cells respond to immune cell attacks by expressing *IDO1* and recruiting CCL22-positive cells (Fig. 6e). Notably, even in an immunosuppressive environment, some CD8 + T cells still expressed *GZMB* and *PRF1*, indicating that T-cell cytotoxicity was active. Thus, activating the remaining immune cells with ICBs could result in elimination of malignant-transformed mucinous tumor cells.

We investigated novel factors for which gene expression was activated by immune cell attacks in the boundary region. To complement the limited number of genes detected in Xenium analysis, we performed reference-free cell-type deconvolution[38] on Visium data (Supplementary Fig. S10a). We extracted 15 cell types. Cell-type X9 was found along the mucinous boundary. Cell-type X9 highly expressed immune response-related genes, such as *IDO1* (log2fc = 2.5). Furthermore, *RHOV* (log2fc = 3.1) was highly expressed in this cell type (Supplementary Fig. S10c). *RHOV* is linked to cell proliferation, migration,

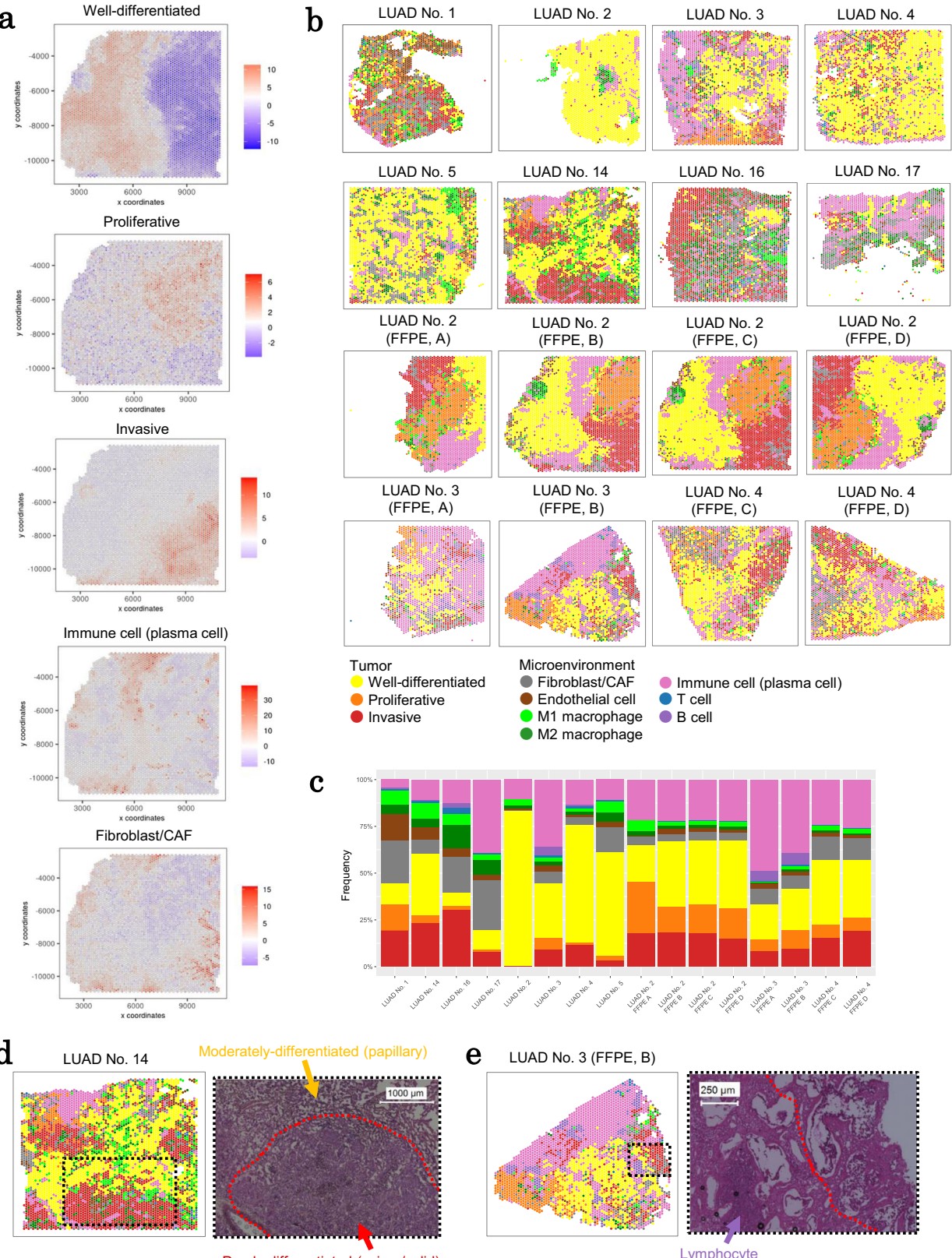

**Fig. 4 | Cross-case or section TME scoring of local expression features. a** TME scoring of four signatures in section C of LUAD No. 2 FFPE. **b** TME scoring is used to determine which features appear in each spot of IA sections. **c** The percentage of each feature of TME scoring in all IA cases. The invasive feature distribution (left) and H&E image (right) for cases LUAD No. 14 in (**d**) and section B of LUAD No. 3 FFPE in (**e**). A dashed box represents the region where the invasive feature is enriched. Source data are provided as a Source Data file for (**c**).

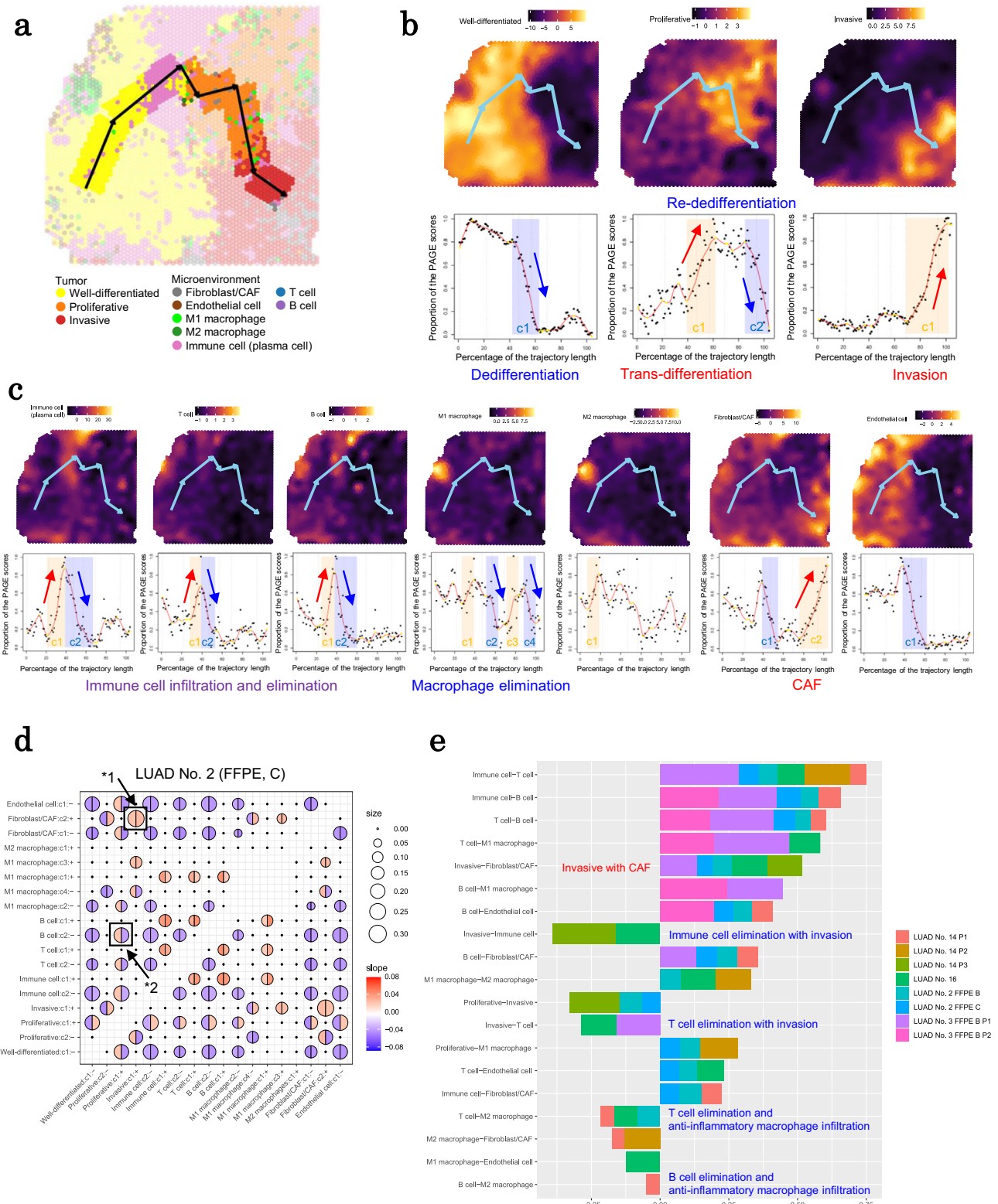

and metastasis in lung adenocarcinoma[39,40]. This observation could be an example of erroneous de- or re-differentiated cells beginning to express a series of genes that promote cancer development.

We assessed the association between tumor cells and CAFs using the improved spatial resolution of Xenium. We investigated the CAF-rich tumor invasion region in LUAD No. 2. In this region, Visium data did not distinguish between cells expressing key genes. For example, *MMP7* and *CXCL14*, which play important roles in tumor cell invasion

and inflammation, respectively, Xenium data clearly indicated that tumor cells were responsible for their expression (Fig. 6f). These findings are critical when considering pharmacological interventions targeting key molecules involved in tumor cell invasion, such as *MMP7* and *CXCL14*[41,42]. On the CAF side, we found expression of other matrix metalloproteinase *MMP11* (Supplementary Fig. S9d). Although we could not identify specific interaction factors between tumor cells and CAFs, we hypothesized that both tumor cells and CAFs (and possibly

**Fig. 5 | Tumor progression with distinct TME landscapes. a** Integrating the transcriptome trajectory with the defined TME features in LUAD No. 2 FFPE section C. The score distribution of each feature and its landscape along the inferred trajectory. Features of tumor cells and their microenvironment are shown in (**b** and **c**), respectively. **d** The positive and negative relationships between TME changes in LUAD No. 2 section C. Each radius represents the length of the tumor cells' evolutional trajectory line, where correlations were found between respective features, as shown on the X and Y axes. The heat color indicates the degree of the slope on the trajectory line for the specific feature. The left and right hemispheres represent the slopes of the features depicted on the X and Y axes, respectively. For example, the circle marked *1 indicates a positive correlation between "Invasive: c1" and "Fibroblast/CAF: c2." Here, the association spanned 0.26 parts of the tumor cells'

evolutional trajectory, with slope ratios of 0.026 and 0.028 for the respective features (both features increased as cancer evolved). The circle marker with *2 represents the anti-correlation between "Proliferative; c2" and "B cell; c2" features. **e** Positive and negative relationships between TME pairs on the eight tumor cell evolutional trajectories of five sections. "cor_st" is the sum of the absolute values of the slopes of the two features. Positive and negative values of "cor_st" indicate that the indicated factors are positively or negatively correlated, respectively. Each color box represents the contribution value of the specified specimen. For more information on the data analysis procedure, see the "Methods" section. The observed key positive and negative correlations are highlighted in red and blue letters, respectively. Source data are provided as a Source Data file for (**d** and **e**).

their engagement) would play a role in ECM remodeling, resulting in the exclusion of anti-tumor immune cells[43–45] from this region.

From the viewpoint of immune environments in invasive regions, anti-inflammatory macrophages are infiltrated, while other cytotoxic immune cells are often excluded. In general, alveolar macrophages expressing *MARCO* are found in lung adenocarcinoma. We compared macrophage profiles to the obtained landscape profiles and found that *SPP1*-expressing macrophages became dominant in moderately differentiated and invasive tumor regions, which is consistent with previous studies[12,16,46,47]. Notably, in LUAD No. 14, macrophages with high *SPP1* expression increased in a region characterized by de-differentiated and invasive tumor cells with acinar and solid histological patterns (Supplementary Fig. S9e). These features were first revealed through single-cell resolution analysis with Xenium.

### Investigation of TME statuses at the earlier phase of tumor progression

Having identified distinct changes in microenvironment statuses and tumor progression in IAs, we sought to investigate whether these changes occurred earlier in tumor development. Therefore, we conducted a similar analysis on very early lung adenocarcinoma cases, such as AISs and MIAs.

Using Visium data (Supplementary Fig. S11), we first assessed TME scoring in local tissue regions of AIS/MIA cases (Fig. 7a and Supplementary Fig. S12), as shown in IA cases. Most of the tumors were "well-differentiated." The immune cell regions were not significantly lower than IAs, indicating that immune responses had already occurred at AIS and MIAs. In contrast, "malignant" regions (the sum of "proliferative" and "invasive" regions) were significantly smaller. Notably, these regions appeared as sparsely distributed "spots." Because the cancer cell trajectory observed in IAs had not yet begun, we had to modify the previously described analytical scheme to characterize the spots in AISs/MIAs. We defined "possibly malignant" regions as those in which a number of proliferative or invasive spots were enriched with a specific enrichment score ("possibly malignant-invasive" or "possibly malignant-proliferative" regions). Overall, the degree of enrichment was greater in IAs for both proliferative and invasive spots, indicating that these regions are more densely packed in IAs (Fig. 7b). Nonetheless, we could depict "possibly malignant" regions from AIS and MIA cases (Supplementary Fig. S13a). In these "possibly malignant" regions, we attempted to characterize TME landscapes.

A total of 25 regions in 11 cases were identified as "possibly malignant" regions in AIS/MIA cases (Supplementary Fig. S13b–d), indicating that AIS/MIAs do contain "possibly malignant" regions, albeit they are typically smaller and sparser. We found that in the "possibly malignant" regions of both AIS/MIA, similar to IA cases, pathways associated with response to various stresses, such as oxidative stress, unfolded protein, growth factors, cytokines, hormone stimulation, and apoptosis, were upregulated, as well as those of invasive phenotypes, such as cell motility and stromal development (Supplementary Fig. S14a–c). Vasculature development signaling was

also increased in "possibly malignant" regions in both AIS/MIA and IA cases (Supplementary Fig. S14a), indicating that endothelial signaling is activated for tumor cell proliferation and invasion beginning in the early stages. These findings indicate that even in AIS/MIAs, core gene expression changes had already begun to progress to IAs.

Nonetheless, distinct gene expressions were found in AIS/MIA versus IA cases. Particularly, genes associated with the existence of stromal and immune cells were highly expressed in some cases (Supplementary Fig. S14a–c). For example, the fibrosis/elastosis-associated genes (*CCN2*, *TIMP3*, *MFAP4*, and *LTBP4*) were found to be overexpressed in "possibly malignant" regions of AIS/MIAs. Among them, the most distinguishing feature was that inflammatory lymphocyte- and/or macrophage-related gene expression was significantly more relevant in the "possibly malignant" regions of AIS/MIAs than in IAs. Immune cells likely infiltrated "possibly malignant" regions in AIS/MIAs more aggressively than IAs (Supplementary Fig. S14d).

A thorough examination of several cases revealed that the more active interaction of immune cells in AIS/MIAs was the true cause. For example, in the "possibly malignant" region of TSU-33, we found inflammatory immune cells, such as B cells (*MS4A1* and *CXCR4*) as well as high levels of the pro-inflammatory chemokine *CCL19* (Fig. 7c). This "possibly malignant-invasive" region contained several lymphocyte-enriched structures, which we confirmed using PhenoCycler analysis (see our report[16]). Similarly, M1-like alveolar macrophages were found in the possibly malignant-invasive regions of TSU-19 and TSU-27 (Supplementary Fig. S14e, f). In TSU-30, *FCGR3A* (CD16) was highly expressed in the "possibly malignant-invasive" region, indicating a fibrotic feature associated with high levels of *COMP* and *COL15A1* (Fig. 7d). *FCGR3A*-expressing cells, most likely macrophages, appeared to recruit cytotoxic T lymphocytes. These findings indicate that in early tumors, tumor cells in the interior of "possibly malignant" regions are exposed to inflammatory stimuli from immune cells, which are absent in IAs.

We characterized the immune cell profiles more precisely. As partly described in our previous paper[16], we investigated and found that in the possibly malignant regions of AIS/MIAs, co-localization of FABP4+ and SPP1+ macrophages was especially important (Supplementary Fig. S14g). Macrophages with high *FABP4* expression have been identified as pro-inflammatory, and they are primarily found among normal-like alveolar macrophages[46,48]. These findings collectively suggest that these possibly malignant regions are the regions where tumor cells have just begun to break through the barrier of the immune cells by first transforming their phenotypic appearance.

We then investigated the behaviors of immune cells at a finer resolution using PhenoCycler and Xenium (data summary for all AIS/MIA in Supplementary Figs. S15 and S16; analysis for stromal cells in Supplementary Fig. S17). We found co-localization of high *MARCO*-expressing alveolar macrophages and *SPP1*-expressing macrophages in the "possibly malignant" regions, such as TSU-27 (Fig. 7e). In TSU-30, several *FABP4*-expressing macrophages were found to co-localize with SPP1+ macrophages, even within the same alveolar space (Fig. 7f). Furthermore, we confirmed that lymphocyte infiltration existed in

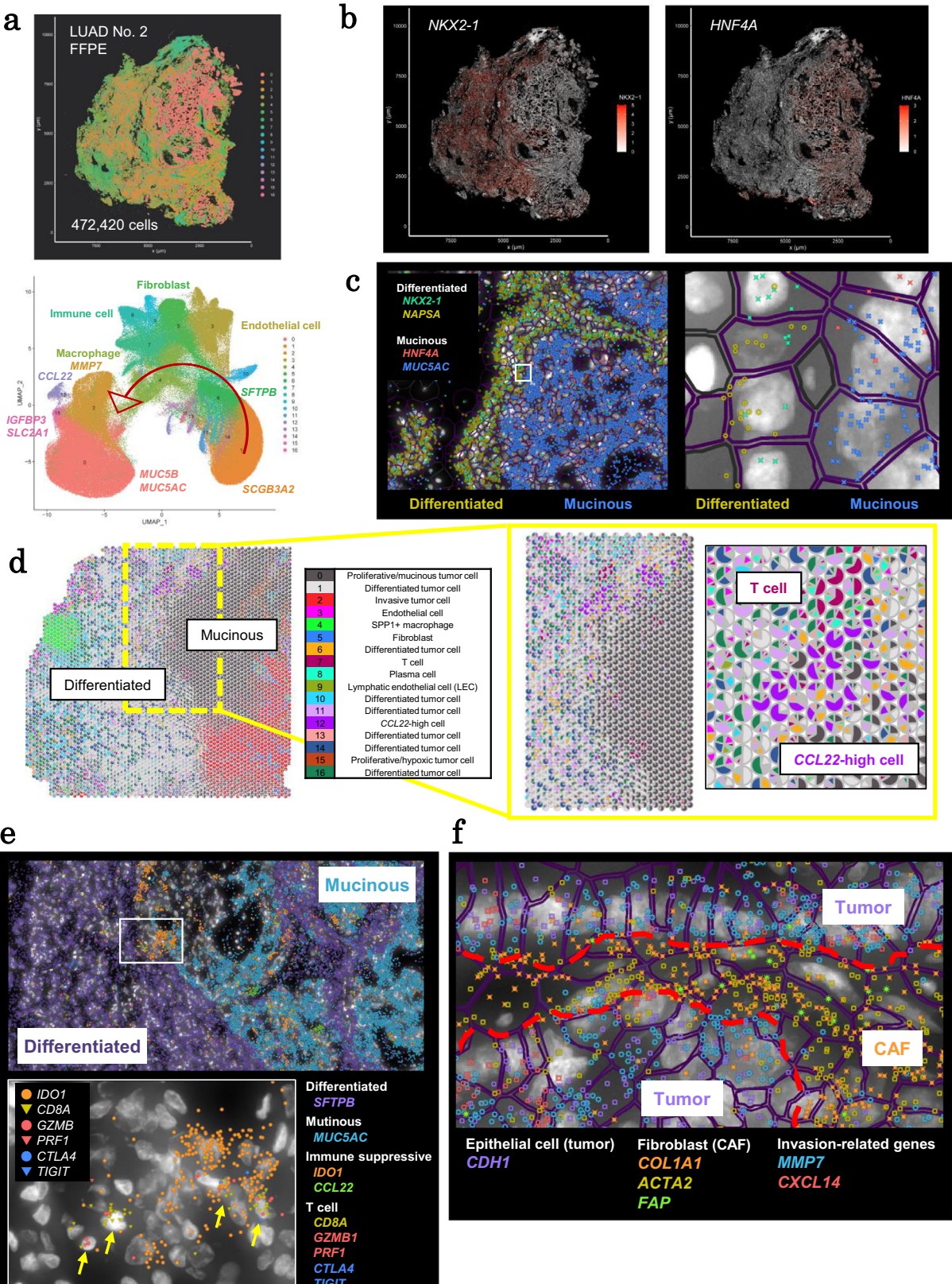

**Fig. 6 | Spatial and single-cell characterization of TME in LUAD No. 2. a** Spatial (upper) and UMAP (lower) plots of Xenium in LUAD No. 2 FFPE. **b** The spatial expression patterns of *NKX2-1* and *HNF4A* in Xenium data. **c** Boundary region of *NKX2-1* and *HNF4A*-positive tumors at the single-cell level. Each plot represents mRNA molecules. **d** Deconvolution analysis of Visium data using Xenium.

**e** Microenvironment statuses of the boundary region between *NKX2-1-* and *HNF4A*-positive regions at the single-cell level. **f** Association between tumor cells and adjacent CAFs at the single-cell level in the invasive region. All image plots in (**a**–**c**, **e**, and **f**) were flipped to match the orientation of Visium.

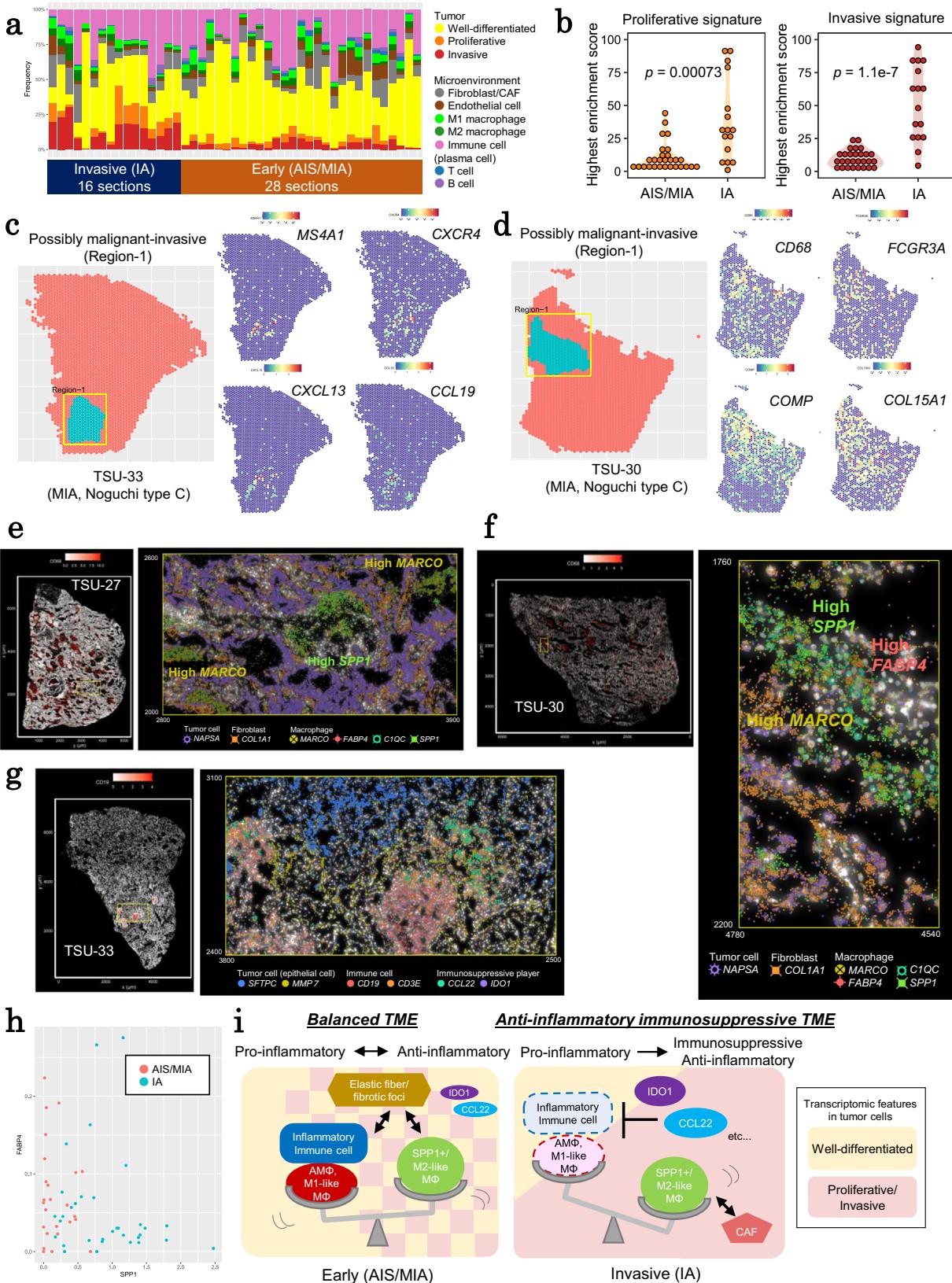

several "possibly malignant" regions, including TSU-33. TLS-like structures with B and T-cell aggregation were observed in "possibly malignant" regions expressing *MMP7* (Fig. 7g). Immunosuppressive cells, such as *CCL22*- and *IDO1*-expressing cells, were found at the boundary between well-differentiated tumors (*SFTPC*) and more invasive tumors (*MMP7*-expressing cells). All of the results support Visium's hypothesis,

which is that, in those "possibly malignant" regions, the interaction between tumor cells and immune cells is changing.

Taken together, in early adenocarcinomas, inflammatory immune subsets can still infiltrate malignant regions (Fig. 7h), where immune cells effectively suppress tumor cell proliferation and expansion. Tumor cells that pass through this initial barrier continue to expand outward

**Fig. 7 | TME characterization of very early cases. a** The proportion of each TME scoring feature across all sections. The data from IA cases were the same as those in Fig. 4c. **b** The highest local enrichment scores of proliferative and invasive signatures in each section were compared between early (AIS/MIA) and IA cases (*n* = 28 and 16 sections from AIS/MIA and IA, respectively). The *p*-values in the inset were calculated using the Wilcoxon rank sum test (two-sided, no multiple comparison adjustments). **c** A "possibly malignant-invasive" region in TSU-33 (left) and genes associated with immune cell existence (right). All four genes showed significantly higher expression in "possibly malignant-invasive" regions. **d** A "possibly malignant-invasive" region in TSU-30 (left) and genes associated with macrophages and fibrosis (right). *CD68* is a macrophage marker. The other three genes showed significantly higher expression in "possibly malignant-invasive" regions. **e, f** The microenvironment statuses of "possibly malignant-invasive" regions of TSU-27 and TSU-30. The spatial expression pattern of *CD68* is shown (left). A plot of RNA molecules of representative genes around the "possibly malignant-invasive" region (yellow in the left panel) is shown (right). Image plots were rotated and flipped to match the orientation of Visium. **g** The microenvironment statuses of the "possibly malignant-invasive" region of TSU-33. The spatial expression pattern of *CD19* is shown (left). A plot of RNA molecules of representative genes in the boundary of differentiated tumors and the "possibly malignant-invasive" region (yellow in the left panel) is shown (right). Image plots were rotated and flipped to match the orientation of Visium. **h** Comparison of *FABP4* and *SPP1* expression in "possibly malignant" regions between AIS/MIA and IA cases in Visium. Each "possibly malignant" region is shown with its average expression levels. **i** A conceptual representation of TME conditions in early (AIS/MIA) and invasive (IA) cases. Source data are provided as a Source Data file for (**a**, **b**, and **h**).

into empty space, where they face harsher immune cell attacks, such as those seen in IA (Fig. 7i). At both early and invasive stages, by inducing drastic changes in cancer cell gene expression, interactions with immune cells play a critical role in determining the fate of cancer cells.

## Discussion

In this study, we demonstrated spatial profiling of lung adenocarcinoma with Visium, Xenium, and PhenoCycler analyses, and we made the rich spatial omics datasets publicly available. We proposed and implemented several scoring schemes to compare regional profiles. In both AIS/MIA and IA cases, we found that immune cells should act as the most significant barrier for cancer cells to initiate invasions or accelerate their expansions. Although this immune barrier may be effective at first, once tumor cells find a way to survive by changing their transcriptome programs, this barrier causes accelerated evolution, triggering the malignant transformation of tumor cells. In fact, we found that this motif is frequently observed at various stages of the tumor in a large number of cases. More precise control over the interaction of tumor cells and immune cells may be the key to better-treating cancer patients at various cancer stages. It should also be noted that, particularly in IAs, even within a single small specimen, tumor cells and TMEs vary significantly. Therefore, the current approach to cancer classification, which relies solely on biomarkers or genomic mutations, may not always accurately reflect the biology of cancer tissues.

One obvious limitation of this study is that it is an observation study, so it was impossible to determine whether the observed phenomenon should be interpreted on a "cause-consequence" or a "simple associated" axis. Strictly, for such an analysis, a validation analysis using an in vitro perturbation analysis should be required. However, analyzing a larger number of clinical specimens may provide additional information. If enriched information on even subtle changes occurring in a small region could be further associated with the overall pathological appearance and clinical outcomes of the patients, it could provide an important clue in identifying truly "causative" events and vice versa. This cycle should lead to more accurate pathological diagnoses. Following that, we believe the next era of cancer discovery and drug development will begin.

## Methods
### Clinical samples

For the IA cases (LUAD series), frozen surgical specimens from eight patients were used. Samples were collected with informed consent at the National Cancer Center Hospital East in Japan. To prepare FFPE tissues, a portion of frozen LUAD No. 2–4 specimens was fixed in 10% neutral-buffered formalin for 24 h and embedded in a paraffin block. For the AIS and MIA cases (TSU series), frozen tissues from 22 patients were used. The institutional ethics committees approved this study at the National Cancer Center, University of Tsukuba Hospital, and the University of Tokyo, Japan. All AIS/MIA samples were collected with informed consent at University of Tsukuba Hospital, Japan. Patient

consent was obtained in writing. Samples from both females and males were used in this study. Sex and gender information was not considered for the statistical analyses of this study because the information was not a focus in this study.

### Spatial transcriptome sequencing using frozen tissues

Frozen tissues embedded in the OCT compound were sectioned to 10 μm thickness and cryosections were placed on a Visium slide (10x Genomics). Fixation, H&E staining, and imaging were performed according to the manufacturer's instructions. Tissue permeabilization was conducted with a 12 min (IA) or 6 min (AIS/MIA) incubation. Following permeabilization of the tissues, cDNA synthesis, amplification, and sequencing library preparation were conducted using Visium Spatial Gene Expression Reagent Kits (10x Genomics). NovaSeq 6000 (Illumina) was used for sequencing.

### Spatial transcriptome sequencing using FFPE tissues

To ensure the quality of tissue blocks, total RNA was extracted using AllPrep DNA/RNA FFPE Kit (Qiagen). The proportion of fragments >200 nt (DV$_{200}$) was measured using a Bioanalyzer (Agilent Technologies) and met the >50 % threshold. Tissue sections at 5-μm thickness were prepared. Deparaffinization, H&E staining, and imaging were performed according to the manufacturer's instructions. Probe hybridization and library preparation were performed using Visium Spatial Gene Expression Reagent Kits for FFPE (10x Genomics) as described in the user guide (CG000407, Rev A, 10x Genomics). NovaSeq 6000 (Illumina) was used for sequencing.

### Multiplexed immunostaining by PhenoCycler

Multiplexed immunostaining was performed using the PhenoCycler system (Akoya Biosciences) following the manufacturer's instructions. Briefly, for FFPE, tissue sections (LUAD No. 2–4) were prepared at 5 μm thickness adjacent to those for Visium analysis using a microtome (HistoCore MULTICUT R, Leica) and placed on a coverslip. The section was deparaffinized, and antigen activation was performed using a pressure cooker for 20 min. Then, tissue sections were stained with 35 antibodies for 3 h (Supplementary Table S4). For FF, tissue sections (TSU-23, 24, 28, 30, 31, and 35) were prepared at 10 μm thickness adjacent to those for Visium analysis using a cryostat (CM1950, Leica) and placed on a coverslip. The sections were fixed and stained with antibodies for 3 h (Supplementary Table S5). For the target panel design, we mainly selected from the representative markers for epithelial cells (E-cadherin or Pan-Cytokeratin), fibroblasts (collagen-IV or α-SMA/ACTA2), endothelial cells (CD31/PECAM1), T cells (CD3e, CD4 or CD8), B cells (CD19 or CD20), and macrophage/DCs (CD68, CD11c). We have also added other molecules identified by the Visium investigation, such as CXCL13 and IDO1. We prioritized the PhenoCycler-inventoried antibodies which were evaluated in multiple types of FFPE tissues by Akoya Biosciences. For custom-conjugated antibodies, we checked the staining quality on test tissues.

The prepared sections were washed and the antibodies fixed. Imaging analysis was conducted using the PhenoCycler instrument (Akoya Biosciences) and BZ-X810 fluorescence microscope (Keyence).

### In situ gene expression analysis by Xenium

Xenium in situ expression analysis was conducted using Xenium Slides & Sample Prep Reagents (PN-1000460, 10x Genomics). FFPE tissues (LUAD No. 2 and 3) were sectioned at 5-μm thickness and placed on the Xenium slide (10x Genomics) according to the manufacturer's instructions (CG000578, Rev A, 10x Genomics). Deparaffinization and decrosslinking of sectioned tissues were performed according to the manufacturer's instructions (CG000580, Rev A, 10x Genomics). FF tissues (LUAD No. 14, 16, and 17, and TSU series) were sectioned at 10-μm thickness using a cryostat and placed onto the Xenium slide (10x Genomics) following the manufacturer's instructions (CG000579, Rev A, 10x Genomics). Fixation and permeabilization were completed (CG000581, Rev A, 10x Genomics).

The Xenium slide was prepared following the user guide for "Xenium In Situ Gene Expression" (CG000582, Rev A, 10x Genomics). To summarize, pre-designed and custom probes were hybridized at 50 °C overnight, washed, ligated, and amplified. Autofluorescence quenching and nuclei staining were performed in the dark. The instrument was run using Xenium Analyzer (10x Genomics). A total of 302 target genes in the pre-designed (202 genes) and custom (100 genes) panels (design ID: 9GT3BT) were provided in Supplementary Data S1[16].

### Computational preprocessing of Visium spatial transcriptome data

Visium raw data was processed with Space Ranger versions 1.2.1, 1.3.0, and 1.3.1 (10x Genomics). Dimensional reduction and clustering analysis for data from IA cases were conducted using Seurat (version 4.0.0)[49]. The Visium trajectory analysis was conducted using Monocle 3 (version 1.0.0)[50,51]. The cluster that highly expressed well-differentiated alveolar epithelial cell markers, such as surfactant protein genes and *NAPSA*, was defined as the root cluster of the trajectory. Module scores were calculated using Seurat AddModule function with the default parameter settings. *TCF7, LEF1, CCR7, SELL, MALL* (for naïve T cells), *CST7, GZMK, GZMA, NKG7, IFNG, PRF1, GZMB, GNLY* (for cytotoxic and effector T cells), *PDCD1, TOX, CXCL13, TIGIT, CTLA4, TNFRSF9, HAVCR2* and *LAG3* (for exhausted T cells) were used to calculate the module score[52]. Ligand–receptor interaction analysis was conducted using COMMOT (version 0.03)[53].

### TME scoring analysis

For TME scoring, signature gene enrichment analysis (Supplementary Table S8a) was performed with PAGE enrichment using Giotto (version 1.1.2)[36]. The score of each signature was computed for each spot and visualized in spatial plots.

### Evaluation of TME changes for IA cases

To assess changes in TME scores for IA cases, the scores were plotted against spatial transcriptome trajectories, as described below (also see Supplementary Fig. S8a and the "Code Availability" section). For the LUAD No. 2 FFPE sections, the four serial sections' merged data was used. 1) Transcriptome trajectories on the UMAP plot were created using Visium data by Monocle 3[51]. The root mode was determined by manually selecting the most normal-like or well-differentiated tumor cluster with high expressions of well-differentiated epithelial markers, such as surfactant proteins. The order of clusters in spatial trajectory paths was manually determined using Monocle 3 results and spatial data (Supplementary Table S8b). 2) The center of each cluster was connected using the createTrajectoryManually function of SPATA2 (version 0.1.0)[54] to project the paths to the spatial plot and draw spatial trajectories. 3) The TME scores on the trajectory were extracted and

smoothed. Furthermore, the inflection points were extracted, and the slopes between them were calculated. 4) The regions with slopes greater than the threshold (Supplementary Table S8b) were extracted as TME changes. 5) To better understand the mutual relationships between TME changes, TME pairs were identified when the regions of each TME change overlapped on the same trajectory.

For Fig. 5e, the scores of association between TME pairs were calculated by adding the absolute values of both slopes (cor_st). When both slopes were positive, the cor_st value became positive (TMEs were positively correlated). When one slope was positive and the other was negative, the cor_st value changed to a negative value (TMEs were negatively correlated). Only TME pairs with positive or negative correlations across all trajectories of all cases are shown (Fig. 5e).

### Evaluation of TME changes for early (AIS and MIA) cases

To evaluate changes in TME scores during tumor progression, particularly in early cancers, malignant regions with proliferative or invasive spots close together were defined as described below (also see Supplementary Fig. S13a and the "Code Availability" section). 1) All spots' neighboring proliferative or invasive spots were counted. The counts were weighted by decreasing them as the distance from the spot increased, and the weighted counts were converted into "local enrichment scores" for each spot. 2) Using only spots with a local enrichment score greater than the threshold, regions with more than 15 spots were classified as "possibly malignant-proliferative" or "possibly malignant-invasive." The threshold was set at 70% of the highest local enrichment score in each section. When this value was less than 10, the threshold was set to 10.

Differentially expressed genes (DEGs) between the "possibly malignant" and other regions were extracted using the Wilcoxon rank sum test with the Seurat FindMarkers function. For this analysis, count data were normalized and log-transformed by Seurat LogNormalize function.

### Computational processing of PhenoCycler multiplexed immunostaining data

Data processing was performed using CODEX Processor (version 1.8). Visualization was performed using the obtained QPTIFF file by QuPath (version 0.3.2)[55]. Cell segmentation was performed using StarDist (QuPath StarDist extension, version 0.3.2)[56] on the QuPath software based on the DAPI signal. The pixel size was set to 0.37 μm with a nucleus expansion of 5 μm. Mean pixel intensities for each marker were calculated as signal intensities in each segmented cell area.

The signal intensity data of PhenoCycler was further analyzed by Seurat (version 5.1.0). Cells were removed with outliers (≥99th percentile or ≤1st percentile) based on the sum of all signal intensities in each cell. The data was normalized using the centered log-ratio (CLR) method across cells (margin = 2). Dimensional reduction was conducted through PCA, and the clustering was performed using the top 10 PCs. Each cluster was annotated using differential marker expressions between clusters. The image plots were rotated and flipped to match the orientation of Visium.

### Image integration of visium and PhenoCycler multiplexed immunostaining data

The image integration procedure was as follows (Supplementary Fig. S6). 1) Mask image generation: for Visium data, the H&E staining image was binarized into black and white after fiducial markers were detected and removed. For PhenoCycler data, images with multiple antibodies (DAPI, CD44, CD4, E-cadherin, β-catenin1, beta-actin, pan-cytokeratin, and Mac-2/galectin-3) were superimposed and binarized. 2) Mask image alignment: the binarized images were scaled, translated, and rotated to maximize image overlap based on intersection versus union. 3) Coordinate transformation: a mathematical function representing transformations used in image alignment was generated to

reciprocally associate pixel positions in Visium and PhenoCycler images. 4) Mapping: signals in Visium and PhenoCycler images were positionally integrated with the mathematical function and its inverse.

For Visium, count data were normalized by SCTransform using Seurat and used as RNA expression levels of each gene. For PhenoCycler, pixels in the QPTIFF images encoded as 8-bit integers (0–255), were used as expression levels of each protein.

### Computational processing of Xenium in situ expression data
The detection patterns of each RNA molecule were visualized using Xenium Explorer (versions 1.1.0, 1.3.0 and 3.0.0, 10x Genomics). Dimensional reduction, clustering, and UMAP visualization were performed using Seurat (version 4.3.0). All images and plots were rotated and flipped to match the orientation of Visium.

### Deconvolution analysis
Deconvolution of Visium expression profiles was performed using Xenium data through RCTD[36] using R package spacexr (version 2.2.1). The cell-type proportions of each spot were plotted using the R package scatterpie (version 0.1.9).

Reference-free cell-type deconvolution was performed for Visium data (LUAD No. 2 FFPE section C) using STdeconvolve (version 1.3.1)[38]. The number of topics was set to 15.

### Statistics and reproducibility
No statistical method was used to predetermine sample size. Sample size was determined by the availability of the specimens. No data were excluded from the analyses. The experiments were not randomized. The Investigators were not blinded to allocation during experiments and outcome assessment. For Visium, 44 datasets from all the 30 cases were analyzed, and one H&E image was obtained in each section. For PhenoCycler, 12 representative cases were analyzed ($n = 1$ in each case). For Xenium, 17 representative cases were analyzed ($n = 1$ in each case).

### Reporting summary
Further information on research design is available in the Nature Portfolio Reporting Summary linked to this article.

## Data availability
Newly obtained sequencing and image data are available in the Japanese Genotype-Phenotype Archive (JGA, http://trace.ddbj.nig.ac.jp/jga), which is hosted by the National Bioscience Database Center (NBDC) and DDBJ with the identifiers JGAS000613 and JGAS000677. Detailed information about the data is also available on the NBDC websites [https://humandbs.dbcls.jp/en/hum0394-v1] and [https://humandbs.dbcls.jp/en/hum0068-v9] for the IA and AIS/MIA projects, respectively. These accession numbers contain raw sequencing and image data. These raw data are available under controlled access due to ethical restrictions because they are defined as personally identifiable information in Japan. Users require the approval to access the data from NBDC (https://humandbs.dbcls.jp/en/guidelines/data-sharing-guidelines) by applying for the data use (https://humandbs.dbcls.jp/en/data-use). The restrictions for granting data are described in the NBDC web page (https://humandbs.dbcls.jp/en/guidelines/security-guidelines-for-users). The processed data is stored in the database DBKERO (https://kero.hgc.jp/)[57] and made freely available on the project's webpage (https://kero.hgc.jp/Ad-SpatialAnalysis_2024.html). Source data are provided with this paper.

## Code availability
The code used in this study is available in the GitHub repositories at https://github.com/asuzuki-asuzuki/Ad-SpatialAnalysis_2024 for basic analyses and TME scoring, and https://github.com/akinaka-dd/LUAD-Spatial for Visium and PhenoCycler image integration.

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

## Acknowledgements

We thank K. Imamura, M. Satake, K. Abe, Y. Kanayama, E. Ishikawa, E. Sekimori, R. Fujinaga, H. Wakaguri, and S. Aoyama for their technical assistance. This work was supported by JSPS KAKENHI grant numbers 22H04925 (PAGS) (to Y.S.) and 21K15566 (to A.S.). This work was supported by the Japan Agency for Medical Research and Development (AMED) P-PROMOTE grant number JP23ama221522 (to A.S.). The supercomputing resource was provided by Human Genome Center, the University of Tokyo (http://sc.hgc.jp/shirokane.html). The computational resource was also supplied by Kashiwa-no-ha Omics Gate (KOG) (https://www.kog.or.jp/en/server.html).

## Author contributions

J.S., M.T., Y.S., and A.S. designed the study. J.S., G.F., J.Z., and A.S. performed experiments. Y.T., G.F., Y. Kuze, Y. Kashima, M.S., A.K., A.N., and A.S. contributed to the analysis of sequencing and image data. J.S., K.N., H.K., D.M., M.N., and M.T. coordinated the samples. J.S., H.K., Y.N., M.K., D.M., M.N., and G.I. performed tissue preparation and histopathological evaluation. S.N., K.T., M.S., A.K., T.K., M.T., and Y.S. interpreted the findings and supervised the study. Y.T., Y.S., and A.S. wrote the manuscript. All authors have approved the final version of the manuscript.

## Competing interests

The authors declare no competing interests.
