## [Peer Review File · Nature Communications]

REVIEWER COMMENTS

Reviewer #1 (Remarks to the Author): Expert in single-cell and spatial omics, cancer genomics, and tumour microenvironment

This study investigates the dynamic interactions between tumor cells and the tumor microenvironment in lung adenocarcinoma through spatial RNA profiling of samples from different progression stages (non-invasive and invasive). Utilizing advanced sequencing techniques, the authors analyze spatial transcriptome data integrated with in situ RNA profiling from 30 patients. It highlights how interactions with immune cells lead to phenotypic changes in tumor cells, facilitating their progression and evasion through complex cellular mechanisms. This is a comprehensive but mostly descriptive study that could serve the community as a resource with a rich and well-annotated dataset.

The manuscript in its current form overstates several findings by implying causative links and is insufficient in terms of data accessibility (only closed repositories for a subset of the data) and lacks any reported code to ensure reproducibility of the findings (which is unacceptable for a mostly computational paper and invalidates the resource aspect to the community). These issues can be fixed in a revised version in the opinion of this reviewer.

Main points:

The results section is rich with data and provides a detailed account of the spatial and phenotypic heterogeneity observed in tumor samples. However, the interpretation of these data leaps to conclusions without sufficient evidence from the data presented. The authors claim a causative link between phenotypic changes and interaction with immune cells (abstract and introduction). I am not convinced that the proposed analyses are sufficient to claim causation over correlation. However, in the way results and conclusions are presented, the causal link is not as strong and doesn't seem to be the ultimate focus of the paper. The authors should avoid using words such as "cause" "trigger", and "causative event" in the abstract and introduction or provide functional evidence with perturbation experiments to prove a causative link.

Evaluation of TME changes: I think it is rather unclear how they integrate the trajectory analysis (code lacking) into the TME feature scoring (selection of the spots for TME score extraction). Moreover, the authors report both using Spata as well as Monocle 3 for trajectory inference. It is unclear how these two tools were integrated into their pipeline (again, code lacking).

The method section of this computational study is sparse and does not enable the reproduction of the results. It is crucial that a well-annotated version of all the utilized code and all the underlying image and sequencing data are published in an easily accessible repository before further consideration of the study.

Data access: The currently provided data access links are insufficient (require prior authorization of a committee, no peer review possible). At least all of the processed sequencing data and all of the imaging and Xenium data are not identifiable and have to be provided in an appropriate open repository (GEO,

Zenodo.), the corresponding documentation has to enable reproduction of all the reported results with the deposited code.

Minor points:

108-111: since PhenoCycler and Xenium analyses were performed for a subset of samples only, it would be clearer to rephrase this. EG. "Some of the interactive events" or "Interactive events for some representative samples". I think this would help the reader to know what to expect while reading the following sections.

123: It would be more accessible to have information reproduced in this manuscript in the form of a supplementary table as well, for consistency and ease.

145: the authors should show violin plots for this (Duox1) and the other mentioned markers (COL1A2, ACTA2, SPARC) for consistency. In the case of these last 3 markers, a violin plot with an average gene expression signature could be shown

Fig 2f description in text is unclear, please specify what "right peripheral" refers to (whole upstream region or subregion within the shown excerpt window).

313: How was this association confirmed? Statistical tests, and if so, which one?

322: I think it's not really clear what "profiling score planner" refers to here. I would use the same terminology used above, such as "transcriptomics statuses scoring"

852: The authors should explain the unit of the plot axis. The X-axis represents the spatial dimension along the slice, I assume it is in the percentage of the trajectory length (but not specified).

Reviewer #2 (Remarks to the Author): Expert in lung adenocarcinoma genomics, progression, evolution, computational analysis, tumour microenvironment and immune microenvironment

The study by Takano et al. uses the latest spatial technologies to profile a valuable clinical cohort of early and invasive lung adenocarcinomas. They discover that there is phenotypic heterogeneity across a single specimen and also between different histology blocks from a single tumour. They find that local areas of immune infiltrate are anti-correlated with tumour aggressiveness and posit that these immune responses suppress locally invasive tumour phenotypes. The data appears to be of high quality and methodologically sound, although additional methodological details would be required.

Despite the use of interesting clinical samples and advanced technologies, this study requires a major rework to improve comprehension. At present, it is challenging to take away key learnings from this paper.

My review points are below.

Major comments

1. From the beginning, the clinical cohort and use of technologies is hard to follow. The authors include Table 1 and Supplementary Table 1 and Overview of study design, which describes the cohort and the technologies applied. However, there is no rationale explaining why some samples (e.g. LUAD No. 2) have 4 serial sections profiled by Visium while other cases only have 1 section. Similarly, the overarching rationale for selecting some cases for phenocycler/Xenium profiling is missing. Was the tissue exhausted, or were only cases of interest followed up based on biological questions?

Why were some samples profiled from FF and some from FFPE?

I believe a study overview schematic in Figure 1 would provide much more clarity on the cohort and the technologies used. I would encourage including details such as the number of transcripts/proteins analysed and the resolution (microns) by each platform.

2. While I am sympathetic to the challenges of distilling such rich data into narratives, I found the results section confusing. Please can the authors pair each statement with a specific figure reference, referring to figure panels rather than a multi-page supplemental figure? More generally, I think the results should be revised and more focused. The data is definitely interesting and could be very impactful if repackaged into a clearer story.

3. Why are only LUAD No. 2 and LUAD No. 3 selected as the focus for the first Results sections? The study overview sets the stage for a broader analysis and then it seems as though the data from 6 of the 8 patients is overlooked. I recognise that this data comes back later on, but without the proper set up it was confusing jumping in and out of focused analyses.

4. It is interesting in LUAD No. 2 that the tumour-stromal interface rich with CAFs is more proliferative and lower in immune cells. Can the authors infer which scenario is more likely - tumour cell engagement with stroma which pushes out immune cells, or tumour cell adaptations that limit immune cell engagement leading to filling in of the stromal areas with CAFs/collagen/ECM?

5. Is there any indication that hypoxia/endothelial signalling/spatial localisation is important in regulating progression from AIS/MIA to invasive disease?

6. The term 'PM – possibly malignant' is confusing. I do not believe it is a standard acronym and could also be read as 'pre-malignant'. I would suggest the authors select another term.

7. Possibly as there is no held-out Discussion section, there is a tendency to over-state potential

interpretations of the data at the end of each results section. In general, I noticed throughout that there are several statements where a reference was lacking. The manuscript requires a careful review of claims, including citations where needed, and a Discussion section to place the findings in the context of the literature. There are some very interesting findings in this manuscript related to tumour progression, but currently the true implications are not obvious.

Methods

1. Was there high concordance between serial sections profiled by the same technology? This goes back to the rationale for using 4 serial sections for Visium LUAD No. 2.

2. How was the phenocycler panel developed and how was the quality of staining evaluated/ensured? Insufficient methodological details were provided to explain how cell segmentation and phenotyping were performed and evaluated.

3. It is clear that this study has benefitted from the expertise of pathologists. Their involvement in case review and interpretation of data could be expanded upon.

Minor comments

1. The manuscript requires a grammatical review.

2. Figure 1: it is difficult to determine alignment of representations from visium (a) and Phenocycler (f). It would help to have these defined as squares in the matching H&E.

3. Figure 2: The heatmap is quite large and does not provide much information. I would suggest saving space in the main figure for panel f which is currently difficult to read.

4. Supp Figure 14: the p-value vertical lines are not defined.

5. Line 84: Could the authors please add a reference related to cytotoxic CD8 T cell localisation to TLS.

6. 171-172: It is not clear where the lymphoid follicle is localised in Supp Fig 4a.

Point-by-point responses to Reviewer #1:

Reviewer #1 (Remarks to the Author): Expert in single-cell and spatial omics, cancer genomics, and tumour microenvironment

This study investigates the dynamic interactions between tumor cells and the tumor microenvironment in lung adenocarcinoma through spatial RNA profiling of samples from different progression stages (non-invasive and invasive). Utilizing advanced sequencing techniques, the authors analyze spatial transcriptome data integrated with in situ RNA profiling from 30 patients. It highlights how interactions with immune cells lead to phenotypic changes in tumor cells, facilitating their progression and evasion through complex cellular mechanisms. This is a comprehensive but mostly descriptive study that could serve the community as a resource with a rich and well-annotated dataset.

The manuscript in its current form overstates several findings by implying causative links and is insufficient in terms of data accessibility (only closed repositories for a subset of the data) and lacks any reported code to ensure reproducibility of the findings (which is unacceptable for a mostly computational paper and invalidates the resource aspect to the community). These issues can be fixed in a revised version in the opinion of this reviewer.

We appreciate the reviewer's careful evaluation. We agree that some parts are overstated, and toned down in this revision, as described below. Regarding data access, we have opened the DDBJ dataset. At the same time, we made the spatial analysis datasets available to the public via our own database. All of the codes have also been made available through GitHub.

- Data (our database)
https://kero.hgc.jp/Ad-SpatialAnalysis_2024.html
- Code (GitHub repositories)
https://github.com/asuzuki-asuzuki/Ad-SpatialAnalysis_2024
<https://github.com/akinaka-dd/LUAD-Spatial>

For more details, see the below responses.

Main points:

The results section is rich with data and provides a detailed account of the spatial and phenotypic heterogeneity observed in tumor samples. However, the interpretation of these data leads to conclusions without sufficient evidence from the data presented. The authors claim a causative link between phenotypic changes and interaction with immune cells (abstract and introduction). I am not convinced that the proposed analyses are sufficient to claim causation over correlation. However, in the way results and conclusions are presented, the causal link is not as strong and doesn't seem to be the ultimate focus of the paper. The authors should avoid using words such as "cause" "trigger", and "causative event" in the abstract and introduction or provide functional evidence with perturbation experiments to prove a causative link.

We thank the reviewer for carefully reading the manuscript. We completely agree with the reviewer's assessment that we oversimplified the cause-correlation issue. We modified the related descriptions as listed below (for more minor parts, the modified descriptions are highlighted in red in the text).

- **Lines 71–72:** significant changes in the phenotypic appearances of tumor cells were frequently associated with changes in immune cell features
- **Line 114–115:** an important event associated with disease progression

However, as the reviewer mentioned, we believe that these changes should not alter the paper's overall focus. We also included this discussion and our critique of the study in the **Conclusion** section (**lines 517–525**). Again, we appreciate you bringing this important issue to our attention, which may have left readers with a negative impression.

Evaluation of TME changes: I think it is rather unclear how they integrate the trajectory analysis (code lacking) into the TME feature scoring (selection of the spots for TME score extraction). Moreover, the authors report both using Spata as well as Monocle 3 for trajectory inference. It is unclear how these two tools were integrated into their pipeline (again, code lacking).

We apologize for the lack of an explanation on this point. The detailed procedures for the

respective analyses are described in the **Methods** section (line 619–629) and supplementary information (**Supplementary Fig. S8a**) in this revision. To evaluate TME changes in IA cases, we manually determined the spatial trajectory paths (order of the clusters) using Monocle 3 results and spatial information of each spot in Visium data. The paths were then projected onto the spatial plot using SPATA 2 to create spatial trajectories. In terms of code, we have included all of the scripts for TME scoring in the GitHub repository (https://github.com/asuzuki-asuzuki/Ad-SpatialAnalysis_2024). We also included a description of how we evaluated TME changes in early cases (AIS and MIA) in **Supplementary Figure S13a**, as well as the codes in the same GitHub repository.

The method section of this computational study is sparse and does not enable the reproduction of the results. It is crucial that a well-annotated version of all the utilized code and all the underlying image and sequencing data are published in an easily accessible repository before further consideration of the study.

We once again thank the reviewer for bringing this important issue to our attention. In terms of codes and procedures, we improved the descriptions in the text and supplementary files. All images and sequencing data are publicly (anonymously) available through DDBJ and our website. In fact, to facilitate data access, we have begun discussions with the staffs of DDBJ (as well as dbGaP and EGA, through the international framework of INSDC). Particularly for image-based data, such as that of Xenium and PhenoCycler, currently, there is no appropriate platform for the data repository (note that this issue is also related to how much the image data is associated with the patient's personal information; the open access level would differ accordingly). However, once DDBJ had implemented such a portfolio, we will integrate all of the datasets there.

Data access: The currently provided data access links are insufficient (require prior authorization of a committee, no peer review possible). At least all of the processed sequencing data and all of the imaging and Xenium data are not identifiable and have to be provided in an appropriate open repository (GEO, Zenodo.), the corresponding

documentation has to enable reproduction of all the reported results with the deposited code.

We apologize for this inconvenience. This is related to the national policy on data sharing. Even for processed sequencing and image data, it is still “gray” to deposit them for free access in a national database, DDBJ. Please understand that changing the scheme and related documentation takes time. For this part, please give us a little more time. If you have any concerns about this issue, please contact the DDBJ staffs (perhaps through the office of Nature Communications). We deposited all of the data in the public database at the JGA of NBDC/DDBJ with controlled access. We believe this should be considered “public data sharing,” albeit in a conservative manner (Japan’s ethical guidelines prohibit anonymous reviewers from accessing controlled access data). To ensure that the data has been properly deposited, we have included links to web pages for JGAS000613 (IA) and JGAS000677 (AIS/MIA) below;

<https://humandbs.dbcls.jp/en/hum0394-v1>

<https://ddbj.nig.ac.jp/resource/jga-study/JGAS000613>

<https://humandbs.dbcls.jp/en/hum0068-v9>

<https://ddbj.nig.ac.jp/resource/jga-study/JGAS000677>

To allow for easier access, we posted the processed data in our database. The dataset includes all Visium and Xenium image files, we well as count matrices. The data is freely available without requiring any authorization (https://kero.hgc.jp/Ad-SpatialAnalysis_2024.html), and the URL is listed in the **Data Availability** section.

Minor points:

108-111: since PhenoCycler and Xenium analyses were performed for a subset of samples only, it would be clearer to rephrase this. EG. "Some of the interactive events" or "Interactive events for some representative samples". I think this would help the reader to know what to expect while reading the following sections.

In response to the reviewer’s suggestion, we changed the sentence to “Interactive events for some representative samples” in the revised manuscript (lines 107–108).

123: It would be more accessible to have information reproduced in this manuscript in the form of a supplementary table as well, for consistency and ease.

We added the driver gene as well as clinic-pathological information to **Supplementary Table S1 (line 122)**.

145: the authors should show violin plots for this (Duox1) and the other mentioned markers (COL1A2, ACTA2, SPARC) for consistency. In the case of these last 3 markers, a violin plot with an average gene expression signature could be shown

We provided violin plots for *DUOX1* and the other mentioned markers (**Supplementary Fig. S4c (lines 152–153)**) and highlighted the cluster(s) with higher expressions of these genes in red triangles in the margin. We also calculated the expression signatures of three fibroblast markers using Seurat AddModuleScore function to display their average expression signature. We also included a violin plot of the signature in **Supplementary Figure S4d (lines 157–158)**.

Fig 2f description in text is unclear, please specify what “right peripheral” refers to (whole upstream region or subregion within the shown excerpt window).

We apologize for the incorrect explanation. The focused region is the left peripheral of cluster 5 (the entire “downstream” region). We changed the text to “*the left peripheral region of cluster 5 (“Downstream” in Fig. 3e)*” (**lines 231–232**) and titled “Upstream” and “Downstream” to each enlarged images in **Figure 3e** in the revised manuscript.

313: How was this association confirmed? Statistical tests, and if so, which one?

We apologize for the confusing description. We simply manually examined the association in **Figure 4b** to generate hypotheses. We changed “confirmed” to “manually

examined” in this sentence (line 323).

322: I think it's not really clear what "profiling score planner" refers to here. I would use the same terminology used above, such as "transcriptomics statuses scoring"

We decided to use the terminology “TME scoring” (line 335).

852: The authors should explain the unit of the plot axis. The X-axis represents the spatial dimension along the slice, I assume it is in the percentage of the trajectory length (but not specified).

We added the plot axis unit (Fig. 5b and c). The X-axis and Y-axis show the “Percentage of the trajectory length” and “Proportion of the PAGE scores,” respectively. We also added the unit to the plots in **Supplementary Figure S8**.

Point-by-point responses to Reviewer #2:

Reviewer #2 (Remarks to the Author): Expert in lung adenocarcinoma genomics, progression, evolution, computational analysis, tumour microenvironment and immune microenvironment

The study by Takano et al. uses the latest spatial technologies to profile a valuable clinical cohort of early and invasive lung adenocarcinomas. They discover that there is phenotypic heterogeneity across a single specimen and also between different histology blocks from a single tumour. They find that local areas of immune infiltrate are anti-correlated with tumour aggressiveness and posit that these immune responses suppress locally invasive tumour phenotypes. The data appears to be of high quality and methodologically sound, although additional methodological details would be required. Despite the use of interesting clinical samples and advanced technologies, this study requires a major rework to improve comprehension. At present, it is challenging to take

away key learnings from this paper.

We thank the reviewer for carefully reading the manuscript. In this revision, we present detailed information about the procedure. For more conceptual issues, we agree that, even with cutting-edge technology and carefully chosen material, drawing solid conclusions at this stage is not always easy. In our opinion, this is primarily because reconstructing interactions between cancer cells and immune cells *in vitro* remains difficult. Without such a validation analysis, we cannot distinguish between “causative” and “associated” events (also see the response to Reviewer #1). Nonetheless, we believe that this paper contains important information that should be analyzed.

My review points are below.

Major comments

1. From the beginning, the clinical cohort and use of technologies is hard to follow. The authors include Table 1 and Supplementary Table 1 and Overview of study design, which describes the cohort and the technologies applied. However, there is no rationale explaining why some samples (e.g. LUAD No. 2) have 4 serial sections profiled by Visium while other cases only have 1 section. Similarly, the overarching rationale for selecting some cases for phenocycler/Xenium profiling is missing. Was the tissue exhausted, or were only cases of interest followed up based on biological questions?

Again, we appreciate the reviewer’s careful reading of this manuscript. Based on the reviewer’s comments, we included an overview of the study in **Figure 1**, including the descriptions of analytical platforms, samples (cohort), and methods. Also for the purpose of the data sharing, we included all of the datasets, but as the reviewer suggested, the purposes of the data collections vary. For examples, in LUAD No. 2, we wanted to compare the results of FF (fresh frozen) and FFPE samples. In some cases, specimens were collected only when either preparation was available. In fact, in Visium analysis, the experimental protocols differ significantly for FF and FFPE. Both procedures have advantages and disadvantages. FFPE has better tissue morphology and histology than FF. However, when it comes to biochemical assays, such as full-length RNA sequencing

and epigenome analyses, FF samples are far more suitable. In FFPE samples, RNA is generally degraded, so a probe-based approach is used to capture transcriptome profiles. For another example of the study design, for some specimens, several sections were prepared to test the reproducibility of data collection (using consecutive sections) and the horizontal spread of the profiles (using adjacent sections). Because these are technical details and the text is already too long, we described this section in the supplementary file (the footnote of **Supplementary Table S1**). This revision includes the previously mentioned information as well as additional details. More precisely, in **Supplementary Figure S3a–e**, we confirmed the reproducibility of four serial sections in LUAD No. 2 FFPE. We also attempted a combined analysis of the FF and FFPE sections of LUAD No. 2 (**Supplementary Fig. S3f**).

Sample selection was made initially for LUAD No. 2 and 3, simply because they have typical driver mutations (*KRAS* and *EGFR*, respectively) and exhibited diverse local expression profiles. Those specimens were then chosen for the PhenoCycler and Xenium analyses because we believed that the interactions between the presumed cell types, based on the marker expression patterns in the Visium analysis, should be confirmed at the single-cell level. For other cases, we prioritized cases with both well-differentiated and malignant expression features that were more heterogeneous. It is also true that, in some cases, we could not conduct a thorough analysis because the remaining tissues had been exhausted (e.g., we had no remaining tissues of LUAD No. 4 FFPE for performing Xenium analysis).

Why were some samples profiled from FF and some from FFPE?

As previously stated, there are some advantages and disadvantages to the respective samples. FF samples are generally superior for measuring RNA profiles because the RNA is intact and not degraded. Non-classical transcripts, such as long non-coding RNAs, are also included. We began this project with FF specimens and conducted Visium analysis on all cases. However, FFPE samples frequently preserve tissue morphology and histology significantly. When we decided to test a hypothesis using morphologically well-characterized regions, precise pathological diagnosis was critical (LUAD No. 2, 3 and 4 FFPE). Therefore, we used the strategy of scrutinizing the FFPE specimens first

to construct a model and then confirming the model with all of the other FF specimen datasets. This description was added to the footnote of **Supplementary Table S1**. We also organized this information in **Figure 1** to help you understand the overall workflow.

I believe a study overview schematic in Figure 1 would provide much more clarity on the cohort and the technologies used. I would encourage including details such as the number of transcripts/proteins analysed and the resolution (microns) by each platform.

We sincerely appreciate the reviewer's suggestion. We included a study overview in **Figure 1**. We also provided the detailed information about each spatial analytical platforms, such as the number of genes analyzed and their spatial resolution. As a result, we believe the outline of the study is clearly represented.

2. While I am sympathetic to the challenges of distilling such rich data into narratives, I found the results section confusing. Please can the authors pair each statement with a specific figure reference, referring to figure panels rather than a multi-page supplemental figure? More generally, I think the results should be revised and more focused. The data is definitely interesting and could be very impactful if repackaged into a clearer story.

We must admit that we did not always successfully represent the data within the limited space of the text. In this revision, we attempted to clarify the relationship between each of the indicated panels and the text. Now, we have reorganized the pairs of a statement and a figure, and most of the statements are associated with each figure panel. So did we with the supplementary figures. Also, in this revision, we attempted to focus more on each of the subthemes in the **Results** section. In fact, one of the goals of this paper is to make this extensive dataset publicly available. We hope that researchers around the world will use this data in new and different ways. We incorporated this discussion into the text (**lines 502–504**).

3. Why are only LUAD No. 2 and LUAD No. 3 selected as the focus for the first Results sections? The study overview sets the stage for a broader analysis and then it seems as though the data from 6 of the 8 patients is overlooked. I recognise that this data comes back later on, but without the proper set up it was confusing jumping in and out of focused analyses.

We appreciate the reviewer's precise criticism. Practically, the stream-line description is intended to be easy to read. We are fully aware that the manuscript is quite long and may not be easy to be read for researchers from various backgrounds, including clinicians. We attempted to address this concern by demonstrating actual example cases before moving to more abstract parts. Precisely, our manuscript is divided into three major sections: 1) developing a hypothesis for mechanisms of IA progression using representative IA cases, 2) conducting an extension analysis of all eight IA cases to confirm the hypothesis of IA progression, and 3) conducting a broader analysis in comparison to the IA and AIS/MIA stage. We included these descriptions of the study overview in **Figure 1**.

More specifically, we focused on LUAD No. 2 and 3 (as step 1 as shown above) for the following reasons. Specifically, those two cases harbored *KRAS* or *EGFR* driver mutations, respectively. *EGFR* and *KRAS* are two of the most important driver genes in lung adenocarcinoma, accounting for a large proportion of cases. Furthermore, FFPE sections of LUAD No. 2 and 3 contained regions with transcriptomic/histological features of both well-differentiated and malignant regions, which is critical for understanding tumor cell and microenvironmental characteristics in boundary regions. Even though those two cases were chosen, we do not believe they are not "extreme" cases. We included these descriptions in the text (**lines 135–141, 145–147 and 206–209**).

4. It is interesting in LUAD No. 2 that the tumour-stromal interface rich with CAFs is more proliferative and lower in immune cells. Can the authors infer which scenario is more likely - tumour cell engagement with stroma which pushes out immune cells, or tumour cell adaptations that limit immune cell engagement leading to filling in of the stromal areas with CAFs/collagen/ECM?

We appreciate the constructive feedback. In LUAD No. 2 (FFPE, section C), CAFs localized with invasive tumor regions (**Fig. 5**) were classified as myofibroblast-like CAFs (myCAF) with high expressions of α -SMA/ACTA2 (**Fig. 2h** and **i**). According to the literatures, tumor cells interact with myCAF, resulting in extracellular matrix (ECM) remodeling and immune cell exclusion (Cords L et al. 2024 *Cancer Cell*, Papait A et al. 2022 *Cancers*, Grout JA et al. 2022 *Cancer Discov*). Actually, tumor cells and CAFs in this region overexpressed matrix metalloproteinases *MMP7*(**Figs. 2i** and **6f**) and *MMP11* (**Supplementary Fig. S9d**), respectively. Although we could not identify specific interaction factors between tumor cells and CAFs, we hypothesized that both tumor cells and CAFs (and possibly their engagement) would play a role in ECM remodeling, resulting in the exclusion of anti-tumor immune cells from this region. We incorporated this discussion into the revised text (**lines 397–402**).

References:

1. Cords L et al. Cancer-associated fibroblast phenotypes are associated with patient outcome in non-small cell lung cancer. 2024 *Cancer Cell* 42(3):396-412.e5.
2. Papait A et al. Fight the Cancer, Hit the CAF! 2022 *Cancers* 14(15):3570.
3. Grout JA et al. 2022 Spatial Positioning and Matrix Programs of Cancer-Associated Fibroblasts Promote T-cell Exclusion in Human Lung Tumors. 2022 *Cancer Discov* 12(11):2606-2625.

5. Is there any indication that hypoxia/endothelial signalling/spatial localisation is important in regulating progression from AIS/MIA to invasive disease?

We appreciate the reviewer's comments. We re-checked the data from that perspective. We discovered that "vasculature development" signaling was upregulated in "possibly malignant" regions in both AIS/MIA and IA cases (**Supplementary Fig. S14a**), implying that endothelial signaling would be activated for tumor cell proliferation and invasion beginning in the early stages. We included this description in the revised manuscript (**lines 442–445**). Actually, in our previous study, we discovered that angiogenesis signaling was upregulated in AIS Noguchi type B compared to more benign Noguchi type A (Haga Y et al. 2023 *Nat Commun*). These signaling pathways would be important even in the early stages of lung adenocarcinoma progression.

We also looked at endothelial cell phenotypes. It had been reported that endothelial cells differentiate into three types, in response to angiogenic signals: migratory tip cells for the guide of vessel sprouts, proliferating stalk cells for extending sprouts, and quiescent phalanx cells which are need to line new vessels (Zeng Q et al. 2023 *Nat Rev Cancer*, Korn C and Augustin HG. 2015 *Dev Cell*). We focused on expression patterns of *PLVAP*, which is a common marker of tumor endothelial cells and expressed in immature stalk cells (Goveia J et al. 2020 *Cancer Cell*). In TSU-33 which harbors two endothelial cell clusters, cluster 9 showed differentially higher expression of *PLVAP* than cluster 2 (**Supplementary Fig. S17**). Cluster 9 cells are located in more invasive regions (lower parts of the tissue section, including the “possibly malignant-invasive” region) while most of cluster 2 cells are in the top part of the tissue where tumors represented transcriptionally/histologically less malignant features (**Supplementary Fig. S17b**), which indicate that cluster 9 cells might be tumor endothelial cells associated with angiogenesis. However, in other cases, endothelial cells widely expressed *PLVAP* (**Supplementary Fig. S17a**). The results indicate that angiogenesis would occur with tumor endothelial cells and begin from early phases of lung cancer cases. Further evaluation will be needed to extend characterization of endothelial signaling in lung cancer progression. We included Xenium images in **Supplementary Figure S17** and added a discussion to the legend.

References:

1. Zeng Q et al. Understanding tumour endothelial cell heterogeneity and function from single-cell omics. 2023 *Nat Rev Cancer* 23(8):544-564.
2. Korn C and Augustin HG. Mechanisms of vessel pruning and regression. 2015 *Dev Cell* 34(1):5-17.
3. Goveia J et al. An integrated gene expression landscape profiling approach to identify lung tumor endothelial cell heterogeneity and angiogenic candidates. 2020 *Cancer Cell* 37(1):21-36.e13

6. The term ‘PM – possibly malignant’ is confusing. I do not believe it is a standard acronym and could also be read as ‘pre-malignant’. I would suggest the authors select another term.

According to the reviewer's suggestion, we decided not to use the acronym "PM." We added "possibly malignant" to the revised manuscript.

7. Possibly as there is no held-out Discussion section, there is a tendency to over-state potential interpretations of the data at the end of each results section. In general, I noticed throughout that there are several statements where a reference was lacking. The manuscript requires a careful review of claims, including citations where needed, and a Discussion section to place the findings in the context of the literature. There are some very interesting findings in this manuscript related to tumour progression, but currently the true implications are not obvious.

We appreciate the reviewer's precise criticism. We initially attempted to separate the **Results** and **Discussion** sections so that it was clear which part was a direct observation and which was a the discussion supported by how much literal evidence. However, we discovered that this separation made the text more difficult to read by separating the introductory discussion that connects the previous and following sections. Instead, in this revision, we attempted to separate the **Results** and **Discussion** within each section. The discussions of each section were compiled in the last paragraph (**lines 194–203, 250–263 and 283–287**).

Methods

1. Was there high concordance between serial sections profiled by the same technology? This goes back to the rationale for using 4 serial sections for Visium LUAD No. 2.

Yes. We confirmed the high concordance of four serial sections of LUAD No. 2 (**Supplementary Fig. S3a–e**). In **Supplementary Figure S3d**, representative markers demonstrated similar expression patterns across sections. In **Supplementary Figure S3e**, scattering plots of expression levels also revealed high correlations between serial sections.

We also performed serial Visium analysis with LUAD No. 3 and 4. We included the results of the comparison analysis for these two cases (**Supplementary Fig. S3g–l**).

2. How was the phenocycler panel developed and how was the quality of staining evaluated/ensured? Insufficient methodological details were provided to explain how cell segmentation and phenotyping were performed and evaluated.

The PhenoCycler panel includes the representative markers for epithelial cells (E-cadherin or Pan-cytokeratin), fibroblasts (collagen-IV or α -SMA/ACTA2), endothelial cells (CD31/PECAM1), T cells (CD3e, CD4 or CD8), B cells (CD19 or CD20), and macrophage/DCs (CD68, CD11c). We have also included other molecules identified by the Visium analysis, such as CXCL13 and IDO1. We prioritized PhenoCycler-inventoried antibodies tested by Akoya Biosciences in multiple FFPE tissues. For custom-conjugated antibodies, we tested the quality of staining using test tissues (appendix, etc.). We added these details to the **Methods** section (**lines 568–576**).

For cell segmentation, we used StarDist via stardist-extension of QuPath. We included the description and parameters in the **Methods** section (**lines 663–665**). To conduct a more comprehensive phenotyping analysis of PhenoCycler data, we also conducted clustering analysis using signal intensities of all antibodies by Seurat (**Supplementary Figs. S2d–f and S15g–o** for IA and AIS/MIA cases, respectively). We also included the method in the **Methods** section (**lines 666–672**).

3. It is clear that this study has benefitted from the expertise of pathologists. Their involvement in case review and interpretation of data could be expanded upon.

We appreciate the reviewer's constructive suggestion. We work closely with pathologists (note that one of the co-authors, Dr. Ishii (20th) and Dr. Noguchi (17th) proposed a pathological classification of LUAD of the world standard; other pathologists, numbered 6, 10, 11, and 15th in the co-author list, also play important roles in the respective medical institutions). Indeed, this suggestion is significant, as pathologists are becoming increasingly interested in the newly emerging field of spatial biology. We believe that further close collaboration between basic scientists and pathologist will open up new avenues for cancer research. We included the discussion in the text (**lines 521–525**).

Minor comments

1. The manuscript requires a grammatical review.

We have submitted the revised manuscript for English proofreading.

2. Figure 1: it is difficult to determine alignment of representations from visium (a) and Phenocycler (f). It would help to have these defined as squares in the matching H&E.

To make each region more understandable, we defined regions-of-interest ROI-1 and ROI-2 for **Figure 2c–f** and **2g–i**, respectively, in the revised manuscript. We added squares to the appropriate regions (**Fig. 2a**), we well as the descriptions to the figure legends.

3. Figure 2: The heatmap is quite large and does not provide much information. I would suggest saving space in the main figure for panel f which is currently difficult to read.

We moved the heatmap to **Supplementary Figure S5a** and increased the size of panel **f** (new **e**) in **Figure 3** in the revised manuscript.

4. Supp Figure 14: the p-value vertical lines are not defined.

In **Supplementary Figure S14a–c**, enrichment analysis was performed by Metascape (Zhou Y et al. 2019 *Nat Commun*) which calculates p-values using the hypergeometric test. The vertical lines in the graph are included by default in Metascape results, representing gridlines for 2, 4, 6, 10 and 20. We included Metascape's description and citation in the legend. In **Supplementary Figure S14e–g**, differential expressions between the possibly malignant region and other region were assessed by Seurat FindMarkers function using Wilcoxon rank sum test. This description is included in the

legend of **Supplementary Figure S14**.

5. Line 84: Could the authors please add a reference related to cytotoxic CD8 T cell localisation to TLS.

We included the following reference in the revised manuscript (**line 82**).

- Sautès-Fridman C et al. Tertiary lymphoid structures in the era of cancer immunotherapy. *Nat Rev Cancer* 2019 19(6):307–325.
- Fridman, WH et al. B cells and tertiary lymphoid structures as determinants of tumour immune contexture and clinical outcome. *Nat Rev Clin Oncol* 2022 19(7):441–457.

6. 171-172: It is not clear where the lymphoid follicle is localised in Supp Fig 4a.

We included an H&E image of the immune cell-rich region (5. Lymph Follicle) to demonstrate it clearly (**Supplementary Fig. S4a**) in the revised manuscript. Because this region is partially out of frame in section C, we included H&E images of the adjacent section A and confirmed the expression of representative immune cell markers in this region.

REVIEWERS' COMMENTS

Reviewer #1 (Remarks to the Author):

I have reviewed the revised version of the manuscript and have noted significant improvements addressing most of my previous concerns. The authors have enhanced the reporting of their methodology and have facilitated access to some of the supporting datasets. I understand the regulatory issues that prevent data sharing without prior application. However, my personal research experience has led me to conclude that controlled access often impedes academic research due to unreasonable administrative hurdles and timelines. I trust that the authors are interested in making these resources available to the research community and will further advocate for policy changes accordingly (i.e., a reasonable process for controlled-access applications and free availability of processed count data and non-identifiable image data). Furthermore, the deposition of the processed data at https://kero.hgc.jp/Ad-SpatialAnalysis_2024.html is commendable. I trust the authors will maintain these resources and will not close their private repository upon publication of the manuscript. I recommend that, in the future, the authors use a permanent repository for data archiving that does not depend on their own maintenance (e.g., Zenodo). I appreciate the deposition of the code used in GitHub. The authors should create a permanent repository by additionally linking the GitHub repository with Zenodo (see <https://docs.github.com/en/repositories/archiving-a-github-repository/referencing-and-citing-content>). After clarification of this last issue, I am supportive of the publication of the manuscript.

Reviewer #2 (Remarks to the Author):

I have completed my assessment of the revised manuscript "Spatially resolved gene expression profiling of the tumor microenvironment reveals key steps of tumor cell development in lung adenocarcinoma". I thank the authors for their responses and review of their manuscript. The modification of Figure 1 greatly enhanced understanding of the study up-front, and improved comprehension of the paper. The editing of the text has also improved the flow and appropriately reduced some of the previously perhaps overstated conclusions. I thank the authors for their edits to the Figures and additional methodological details. I am satisfied with these changes, and believe this study provides intriguing insights into the field. I believe the data will be an excellent resource for the community.

I have minor comments:

Lines 147 and 207: I would suggest you reference Table 1 after the mention of common EGFR/KRAS mutations. Some readers may want to know where to quickly check for their driver of interest.

Line 260: I would recommend downplaying the language around the conclusion here, perhaps stating 'it is more likely that ICB efficacy would be limited.' One cannot predict phenotypic shifts that would occur on ICB.

Lines 283-287: I would suggest adding a concluding sentence to this paragraph to finish the thought.

Line 378: I suggest changing “can” to “could result in elimination of”.

Line 495: I suggest rewording “out space” to “expand outward into empty space”.

Line 525: I suggest changing “These cycle” to “This cycle” and removing the repetition of the word “cycle” from the start of the next sentence.

Reviewer #2 (Remarks on code availability):

I reviewed the documentation on github which had sufficient documentation in my opinion, but did not have time to test the code myself.

Point-by-point responses for the 2nd revision

Point-by-point responses to Reviewer #1:

Reviewer #1 (Remarks to the Author):

I have reviewed the revised version of the manuscript and have noted significant improvements addressing most of my previous concerns. The authors have enhanced the reporting of their methodology and have facilitated access to some of the supporting datasets. I understand the regulatory issues that prevent data sharing without prior application. However, my personal research experience has led me to conclude that controlled access often impedes academic research due to unreasonable administrative hurdles and timelines. I trust that the authors are interested in making these resources available to the research community and will further advocate for policy changes accordingly (i.e., a reasonable process for controlled-access applications and free availability of processed count data and non-identifiable image data). Furthermore, the deposition of the processed data at https://kero.hgc.jp/Ad-SpatialAnalysis_2024.html is commendable. I trust the authors will maintain these resources and will not close their private repository upon publication of the manuscript.

I recommend that, in the future, the authors use a permanent repository for data archiving that does not depend on their own maintenance (e.g., Zenodo).

I appreciate the deposition of the code used in GitHub. The authors should create a permanent repository by additionally linking the GitHub repository with Zenodo (see <https://docs.github.com/en/repositories/archiving-a-github-repository/referencing-and-citing-content>).

After clarification of this last issue, I am supportive of the publication of the manuscript.

We appreciate the reviewer's encouraging comments. In fact, we totally agree with the reviewer's comments that the present study should be no more than a prelude to further future studies. We appreciate, again, the reviewer for his/her careful reading of our manuscript.

Point-by-point responses to Reviewer #2:

Reviewer #2 (Remarks to the Author):

I have completed my assessment of the revised manuscript "Spatially resolved gene expression profiling of the tumor microenvironment reveals key steps of tumor cell development in lung adenocarcinoma".

I thank the authors for their responses and review of their manuscript. The modification of Figure 1 greatly enhanced understanding of the study up-front, and improved comprehension of the paper. The editing of the text has also improved the flow and appropriately reduced some of the previously perhaps overstated conclusions. I thank the authors for their edits to the Figures and additional methodological details. I am satisfied with these changes, and believe this study provides intriguing insights into the field. I believe the data will be an excellent resource for the community.

We truly appreciated the reviewer's comments.

I have minor comments:

Lines 147 and 207: I would suggest you reference Table 1 after the mention of common EGFR/KRAS mutations. Some readers may want to know where to quickly check for their driver of interest.

We referred Table 1 in the revised manuscript (lines 147 and 207) according to the reviewer's suggestion.

Line 260: I would recommend downplaying the language around the conclusion here, perhaps stating 'it is more likely that ICB efficacy would be limited.' One cannot predict phenotypic shifts that would occur on ICB.

According to the reviewer's suggestion, we modified the sentence (line 260–261).

Lines 283-287: I would suggest adding a concluding sentence to this paragraph to finish

the thought.

We added the concluding sentence (**line 288–290**) as below;

“To develop therapeutic strategies for such case, it may be better to consider the transcriptomic features and their heterogeneity.”

Line 378: I suggest changing “can” to “could result in elimination of”.

According to the reviewer’s suggestion, we modified the words (**line 381**).

Line 495: I suggest rewording “out space” to “expand outward into empty space”.

According to the reviewer’s suggestion, we modified the words (**line 498**).

Line 525: I suggest changing “These cycle” to “This cycle” and removing the repetition of the word “cycle” from the start of the next sentence.

According to the reviewer’s suggestion, we modified “These cycle” to “This cycle” and removed the “cycle” in the next sentence (**line 527**).

Reviewer #2 (Remarks on code availability):

I reviewed the documentation on github which had sufficient documentation in my opinion, but did not have time to test the code myself.

We appreciate your kind review of the documentation on the Github repository.